# In-situ electrochemical reconstruction and modulation of adsorbed hydrogen coverage in cobalt/ruthenium-based catalyst boost electroreduction of nitrate to ammonia

Jian Zhang [1], Thomas Quast [1], Bashir Eid [1], Yen-Ting Chen [2], Ridha Zerdoumi [1], Stefan Dieckhöfer[1], João R. C. Junqueira[1], Sabine Seisel [1] & Wolfgang Schuhmann [1] ✉

The electroreduction of nitrate offers a promising, sustainable, and decentralized route to generate valuable ammonia. However, a key challenge in the nitrate reduction reaction is the energy efficiency of the reaction, which requires both a high ammonia yield rate and a high Faradaic efficiency of ammonia at a low working potential ($\geq -0.2$ V versus reversible hydrogen electrode). We propose a bimetallic $Co-B/Ru_{12}$ electrocatalyst which utilizes complementary effects of Co–B and Ru to modulate the quantity of adsorbed hydrogen and to favor the specific hydrogenation for initiating nitrate reduction reaction at a low overpotential. This effect enables the catalyst to achieve a Faradaic efficiency for ammonia of $90.4 \pm 9.2\%$ and a remarkable half-cell energy efficiency of $40.9 \pm 4\%$ at 0 V versus reversible hydrogen electrode. The in-situ electrochemical reconstruction of the catalyst contributes to boosting the ammonia yield rate to a high level of $15.0 \pm 0.7$ mg h$^{-1}$ cm$^{-2}$ at $-0.2$ V versus reversible hydrogen electrode. More importantly, by employing single-entity electrochemistry coupled with identical location transmission electron microscopy, we gain systematic insights into the correlation between the increase in the catalyst's active sites and its structural transformations during the nitrate reduction reaction.

As one of the most essential chemicals in modern society, ammonia ($NH_3$) is contributing to fertilizers, pharmaceuticals, and many other industrial applications[1–3]. However, over 96% of $NH_3$ production relies on the Haber–Bosch process, using $H_2$ derived from fossil fuels consuming approximately 1% to 2% of the global energy, and contributing to about 1.5% to the worldwide $CO_2$ emissions[4,5]. With the $NH_3$ market expected to expand significantly due to its potential as clean hydrogen-rich but carbon-free fuel or for storing $H_2$ for transport purposes, alternative green $NH_3$ synthesis routes need to be considered for achieving climate neutrality by 2050[6–8]. Electrosynthetic $NH_3$ formation technologies from nitrogen-containing feedstocks ($N_2$, nitrate, nitrite and nitric oxide) emerge as promising alternatives to the energy- and carbon-emission-intensive Haber–Bosch process due to their milder working conditions and compatibility with renewable energy provision[9–12]. Among those technologies, the electrocatalytic reduction of nitrate ($NO_3RR$) is more thermodynamically favored for

[1]Analytical Chemistry—Center for Electrochemical Sciences (CES), Faculty of Chemistry and Biochemistry, Ruhr University Bochum, Universitsätsstr. 150, 44780 Bochum, Germany. [2]Center for Solvation Science (ZEMOS), Ruhr University Bochum, Universitätsstr. 150, 44780 Bochum, Germany. ✉e-mail: wolfgang.schuhmann@rub.de

producing NH$_3$ compared to the direct N$_2$ reduction reaction (eNRR)[13–18], owing to the lower dissociation energy of the N=O bond (204 kJ mol$^{-1}$)[15,16]. In addition, nitrate sources with concentrations ≥0.1 mol L$^{-1}$ suitable for industrial-scale ammonia electrosynthesis are widely available in wastewater from the fertilizer industries, metal smelters, and nuclear power plants. Thus, NO$_3$RR simultaneously represents a promising waste-to-value strategy that contributes to alleviating the global nitrogen cycle imbalance[9,19–21].

However, the sluggish eight-electron transfer process involved in the NO$_3$RR has led to significant challenges in achieving a highly efficient conversion from NO$_3^-$ to NH$_3$[22,23]. Therefore, developing efficient electrocatalysts is a prerequisite to further promote the application of the NO$_3$RR[24]. In the last 5 years, transition metal-based NO$_3$RR catalysts in alkaline conditions have seen a steep increase in their development, displaying a relatively high FE for NH$_3$ exceeding 75%[25–28]. However, these catalysts tend to require a relatively negative operating potential (around −0.5 to −0.95 V vs. RHE) to achieve a decent NH$_3$ yield rate at a level of 1–5 mg h$^{-1}$ cm$^{-2}$, resulting in high energy consumption. This is mainly attributed to the proton-coupled electron transfer nature of the electrochemical NH$_3$ production, where the adsorbed hydrogen (*H) on the catalyst plays a crucial role in each electron-transfer step[22,29,30]. Most of the transition metal-based catalysts, however, can only supply sufficient *H to maintain a significant surface hydrogenation rate at extremely negative potentials, particularly under alkaline conditions[31]. Recent studies have revealed that catalysts based on transition metals incorporating a Pt-group metal (Ru, Rh, Pd), which exhibits excellent hydrogen adsorption and desorption abilities, can enhance the hydrogenation step at a more positive applied potential (≥−0.4 V vs. RHE), greatly reducing the overpotential for the NO$_3$RR[24,32–35]. However, a subsequent challenge of Pt-group metals is the need for improved suppression of the HER, as high hydrogen coverage on Pt-group metals tends to kinetically favor the HER over the NO$_3$RR[36]. Thus, precisely modulating the appropriate quantity of adsorbed hydrogen by a combination of transition and noble metals is still a challenging endeavor to meet the specific hydrogenation requirements for the NO$_3$RR, ultimately influencing the achievement of a high FE$_{NH3}$.

The NH$_3$ yield rate ($Y_{NH3}$) through electrocatalytic NO$_3$RR is still not comparable to that of the industrial Haber–Bosch process. Therefore, utilizing the full potential of catalysts to further increase the NH$_3$ yield rate and understanding their reaction mechanisms in NO$_3$RR is the key to further promote the application of the NO$_3$RR[37–39]. Exposing more active sites can further enhance the reaction kinetics in the context of the presence of *H on the catalyst surface, ultimately contributing to the high $Y_{NH3}$. Exposing enough active sites is a promising way to counterbalance the restricted reaction driving force arising from the applied low overpotential[40–42], thus contributing to the high $Y_{NH3}$. Employing in-situ electrochemical reconstruction emerges as a highly effective strategy to improve the performance of the catalyst by increasing the number of exposed active sites[43–47], which perfectly aligns with the requirements for attaining elevated $Y_{NH3}$ production in the low overpotential range. Although reconstruction phenomena of catalysts have been extensively reported[48,49], the correlation between an increased number of active sites and in-situ reconstruction remains notably ambiguous, especially in the context of the complex eight-electron transfer NO$_3$RR. This ambiguity can be largely attributed to major interferences by the changing local chemical environment (e.g., local pH value) during the reaction, which cannot be effectively avoided on macroelectrodes[50–54]. The advancement of single-entity electrochemistry (SEE) on nanoelectrodes offers direct insights into the intrinsic activity of single catalyst particles due to the stable local chemical environment during the reaction[55–58]. Additionally, tracking structural changes of the single particle before and after the reaction becomes feasible when paired with identical-location transmission electron microscopy (IL-TEM). Hence,

we consider SEE suitable to contribute directly to the abovementioned correlation.

We designed a bimetallic Co−B/Ru$_{12}$ electrocatalyst by a facile chemical co-reduction method to catalyze the conversion of NO$_3^-$ to NH$_3$. The introduction of Ru effectively addresses the bottleneck problem of insufficient *H arising from using only transition metals, enabling the formed catalyst to initiate the NO$_3$RR in a lower potential range (≥−0.2 V vs. RHE). The inclusion of Co in the catalyst plays a crucial role in balancing the excessive generation of *H from Ru, effectively suppressing the HER and promoting the NO$_3$RR. Consequently, this catalyst exhibits an outstanding FE$_{NH3}$ of 90.4 ± 9.2% and a remarkable half-cell energy efficiency (EE) of 40.9 ± 4% (theoretical value being 43.9%) at 0 V (vs. RHE) in converting NO$_3^-$ to NH$_3$. Simultaneously, the in-situ electrochemical reconstruction in the catalyst contributes to exponentially exposing more active sites during the reaction, thereby boosting the $Y_{NH3}$ to a high level of 7.4 ± 0.6 mg h$^{-1}$ cm$^{-2}$ at 0 V (vs. RHE), and 15.0 ± 0.7 mg h$^{-1}$ cm$^{-2}$ at −0.2 V (vs. RHE), respectively. More importantly, we directly unveil the correlation of continuously increasing active sites and the structural reconstruction of a single Co−B/Ru$_{12}$ catalyst particle on a nanoelectrode. This insight contributes to shedding light on the mechanism of in-situ electrochemical reconstruction during NO$_3$RR and provides guidance for the rational design of more advanced electrocatalysts.

## Results
### Catalyst design and characterization

The Co−B/Ru catalysts were synthesized through chemical co-reduction of Co and Ru ions using NaBH$_{4(aq)}$. Inductively coupled plasma mass spectrometry (ICP-MS) and X-ray photoelectron spectroscopy (XPS) analysis were used to reveal the bulk and surface compositions of the synthesized samples, respectively (Fig. S1, SI). The B content decreased in the Co−B/Ru$_x$ bimetallic system with increasing Ru in both ICP-MS and XPS, may suggest the coexistence of two forms of Co−B and Ru in the bimetallic system. To differentiate between various Co−B/Ru$_x$ bimetallic catalysts, the designation Co−B/Ru$_{12}$ was assigned to the catalyst with a 12% atomic ratio of Ru determined by ICP-MS. Furthermore, as shown in the transmission electron microscopy (TEM) images (Fig. S2, SI), the Ru-containing catalyst exhibits significantly smaller particle sizes compared to pure Co−B, which may provide a higher electrochemical surface area during the NO$_3$RR.

The powder X-ray diffraction (PXRD) patterns (Fig. 1a) show that amorphous Co−B is formed, as evidenced by a broad reflection at 2θ = 45°[59,60], while the Ru patterns unveil the presence of distinct reflections of the Mg-type crystal structure (space group $P6_3/mmc$, $hP2$) of the metallic Ru phase (PDF No 01-077-3315)[61]. The amorphous nature of Co−B and the crystalline property of Ru were further confirmed by high-resolution transmission electron microscopy (HR-TEM) images (Figs. S3 and S4, SI).

In the case of Co−B/Ru$_{12}$, it exhibits a relatively broad peak at 2θ = 43°, which can be ascribed to the (101) and (002) crystal facets of hexagonal Ru. With increasing Ru content in the bimetallic catalysts, the XRD reflections of Co−B/Ru$_{38}$ and Co−B/Ru$_{75}$ are progressively sharpened, indicating an increasing crystallinity. These results imply that the low crystallinity in Co−B/Ru$_{12}$ can be attributed to the increased presence of amorphous Co−B. We further employed selected area electron diffraction (SAED) to gain more insights into the phase constitution of the Co−B/Ru$_{12}$. In Fig. 1b, the SAED ring is distinctly indexable to various facets of hexagonal Ru. Together with the low crystallinity observed in PXRD, this suggests that Co−B/Ru$_{12}$ comprises two distinct phases: one represented by amorphous Co−B and the other by hexagonal metallic Ru.

Electron energy loss spectroscopy (EELS), EDX mapping, and HR-TEM were applied to further determine the distribution of the two phases in Co−B/Ru$_{12}$. Figure 1d shows a uniform distribution of Co and Ru in Co−B/Ru$_{12}$ at the nanoscale, as clearly observed in the EDS

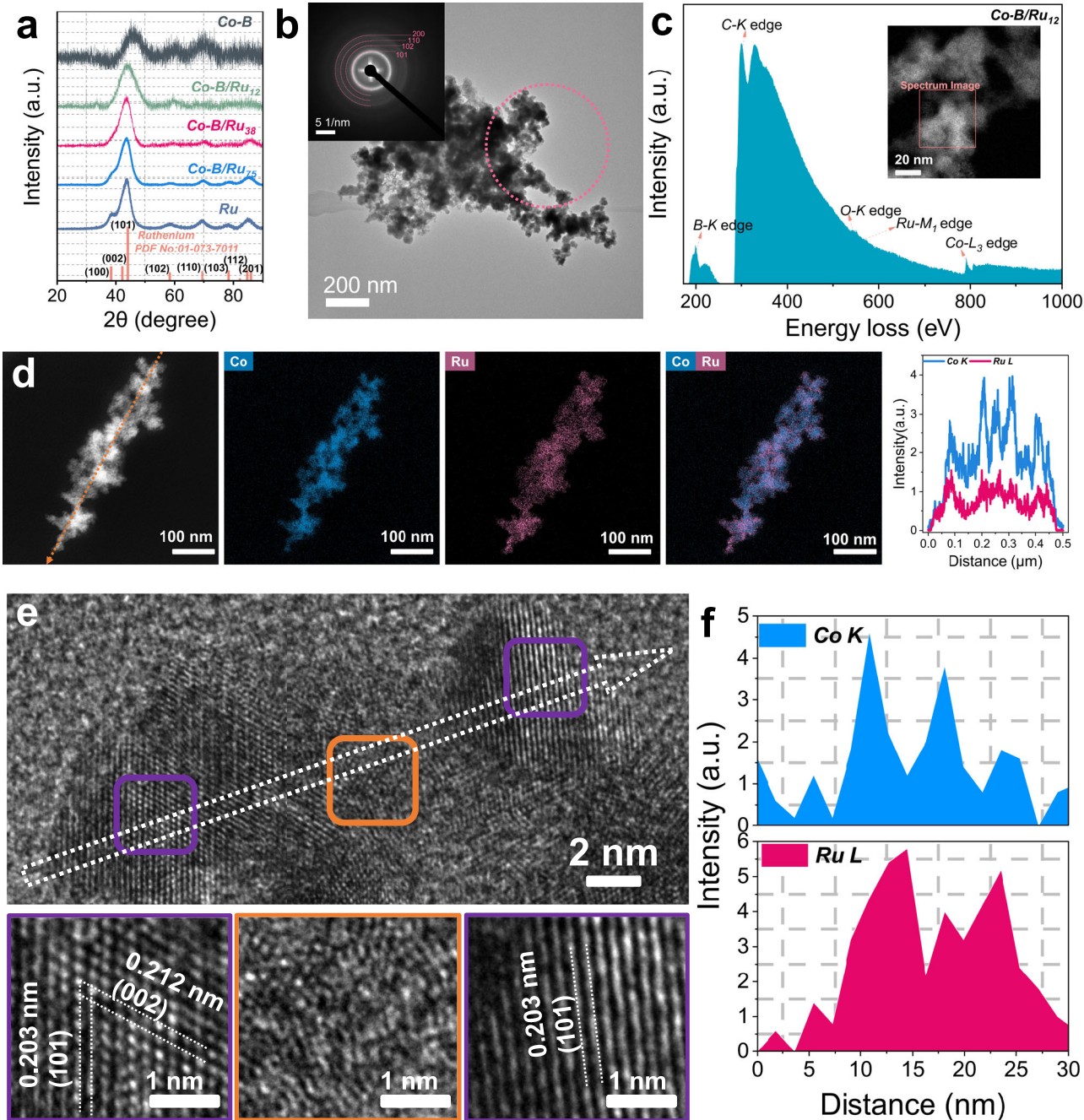

**Fig. 1 | Structural characterization. a** Powder X-ray diffraction (PXRD) patterns of the synthesized catalysts. **b** Transmission electron microscope (TEM) image and selected area electron diffraction (SAED) of Co–B/Ru12. The SAED patterns show the presence of metallic Ru. **c** EELS spectrum of Co–B/Ru12. Backgrounds of zero-loss peak and plasma peaks have been subtracted. **d** EDX element mapping of Co–B/ Ru12 and the corresponding EDX line scan along the designated arrow. **e** HR-TEM image of Co–B/Ru12. The purple-colored rectangle symbolizes the crystalline part, while the orange color signifies the amorphous part. **f** EDX line-scan of Co–B/Ru12 (parallel mode) along the arrow shown in **e**. Source data for this figure are provided as a Source Data file.

mapping and the corresponding EDX line scan. Nevertheless, the electron energy loss spectra further revealed the nanoscale coexistence of Co (L3 edge), B (K edge), and Ru (M1 edge) in the structure of Co–B/Ru12 (Fig. 1c). Furthermore, EELS mapping images in Fig. S5 (SI) shows that B is mixed well with Co. Overall, these results strongly suggest a homogeneous distribution of the two phases in Co–B/Ru12 at the nanoscale. In the HR-TEM images (Fig. 1e), the crystalline region, highlighted in purple, corresponds to the (101) and (002) crystal planes of hexagonal Ru, while the amorphous region, depicted in orange, displays a disordered distribution of atoms. Furthermore, utilizing EDX line scans at the identical location where the HR-TEM

image was taken, we observe an overlap of the two elements along the line (Fig. 1f). This could indicate that Co was integrated into the crystal structure of hexagonal Ru, while Ru is also dispersed within the amorphous Co–B phase.

## Performance of electrocatalytic nitrate conversion

The catalysts were drop-casted onto carbon paper to create uniformly distributed catalyst layers (see "Methods" section for details). Owing to the relatively dense surface structure of the carbon paper (Fig. S6, SI), the drop-coated catalyst was able to uniformly form a 2D film with a thickness of around 5 μm (Fig. 2a and Fig. S7, SI). The NO3RR catalytic

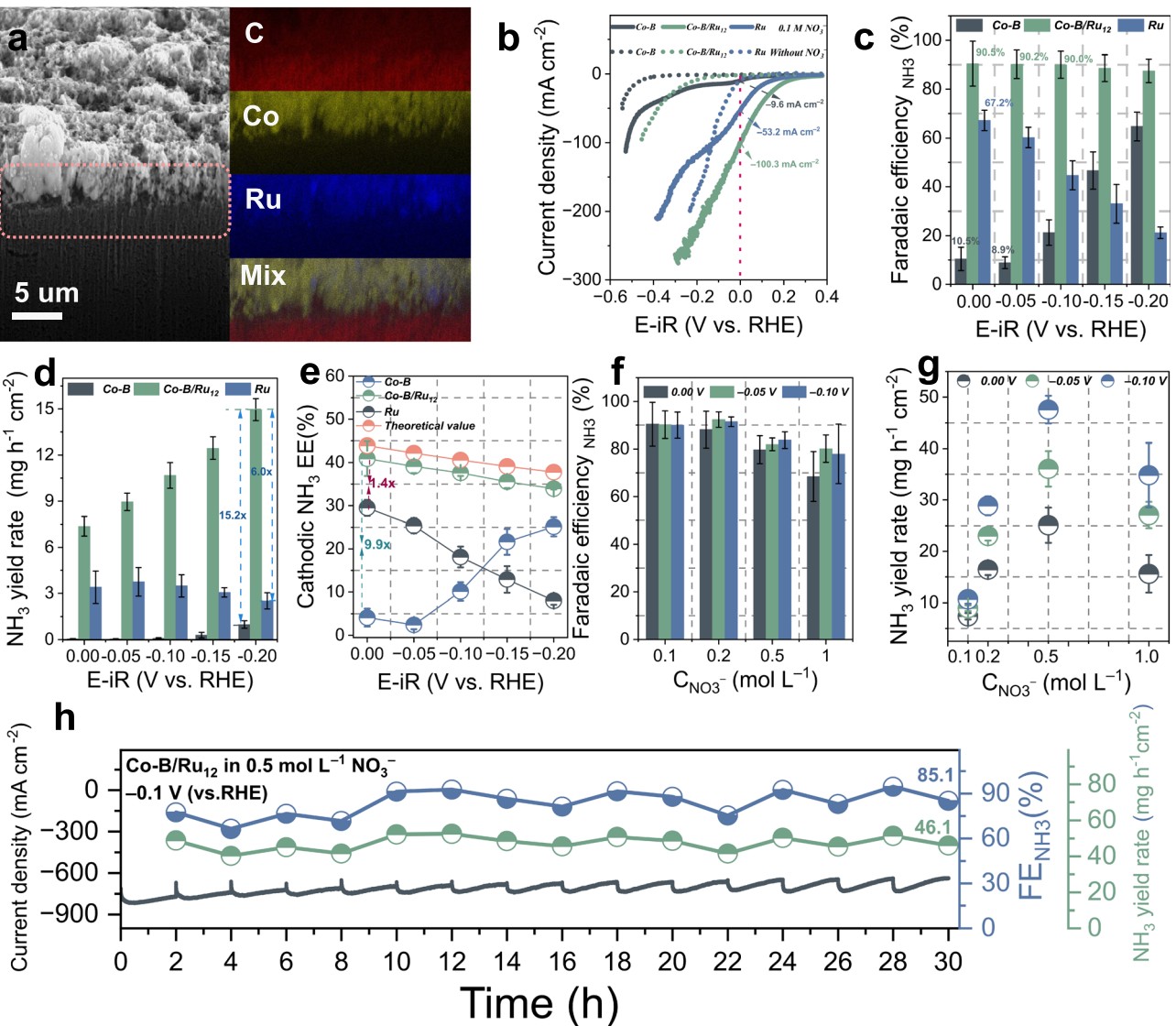

**Fig. 2 | Electrocatalytic NO₃RR performance of Co−B/Ru₁₂, Ru, and Co−B catalysts. a** Focused ion beam-cut scanning electron microscopy (SEM) image and corresponding EDX mapping images of Co−B/Ru₁₂ on carbon paper. **b** Linear sweep voltammograms (LSV) at a scan rate 5 mV s⁻¹ in 0.1 mol L⁻¹ NO₃⁻ and 0.1 mol L⁻¹ NaOH. **c** Faradaic efficiencies of NH₃, **d** yield rate for NH₃, and **e** cathodic energy efficiency for NH₃ on Co−B, Co−B/Ru₁₂, and Ru. **f** Faradaic efficiencies, **g** yield rate for NH₃ on Co−B/Ru₁₂ at various concentration of NO₃⁻. **h** Chronoamperometric stability test of Co−B/Ru₁₂ at −0.1 V (vs. RHE) in 0.5 mol L⁻¹ NO₃⁻, showing the yield

rate for NH₃ and the faradaic efficiencies for NH₃ (with electrolyte replacement each 2 h). The uncompensated resistance, determined by potentiostatic electrochemical impedance spectroscopy, was 18.97 ± 7.69 ohm in 0.1 mol L⁻¹ NaOH, 10.53 ± 2.32 ohm in 0.1 mol L⁻¹ NaOH + 0.1 mol L⁻¹ NaNO₃, 4.78 ± 0.25 ohm in 0.1 mol L⁻¹ NaOH + 0.2 mol L⁻¹ NaNO₃, 4.43 ± 0.36 ohm in 0.1 mol L⁻¹ NaOH + 0.5 mol L⁻¹ NaNO₃, 4.39 ± 0.16 ohm in 0.1 mol L⁻¹ NaOH + 1.0 mol L⁻¹ NaNO₃. Error bars denote the standard deviations from at least three independent measurements. Source data for this figure are provided as a Source Data file.

activity was initially investigated by linear sweep voltammetry (LSV). Unless otherwise noted, all potentials are corrected vs the reversible hydrogen electrode (vs. RHE). The LSV curves (Fig. 2b) of the three samples exhibited significantly increased current densities in the presence of NO₃⁻ compared to the curves obtained in the absence of NO₃⁻. This increase in current density clearly indicates the electrocatalytic activity for the reduction of NO₃⁻. Particularly, Co−B/Ru₁₂ exhibits a lower cathodic overpotential and much higher current density of −100.3 mA cm⁻² (normalized by geometric area) compared to that of Co−B (−9.6 mA cm⁻²) and Ru (−53.2 mA cm⁻²) at a comparatively positive potential of 0 V, illustrating that the enhanced NO₃RR activity can be attributed to the synergistic effect between Co−B and Ru in Co−B/Ru₁₂.

The Faradaic efficiency (FE) for NH₃ of Co−B, Co−B/Ruₓ, and Ru display a significant difference (Figs. S8–S14, SI). Explicitly, as shown in Fig. 2c, Co−B exhibits a negligible FE_NH₃ of 10.5 ± 4.8% and 8.9 ± 2.5%,

respectively, at relatively positive potentials (0 and −0.05 V). At more negative potentials (from −0.05 to −0.2 V), there is a noticeable increase in NH₃ selectivity. In stark contrast, Ru shows a higher FE_NH₃ of 67.2 ± 4.2% at 0 V, yet the FE_NH₃ decreased progressively as the potential became more negative due to the gradually emerging competitive HER. Co−B/Ru₁₂ displays a distinctively improved FE_NH₃ of around 90% (from 0 to −0.1 V) and Co−B/Ru₁₂ maintained its high FE_NH₃ plateau across the applied whole potential range. Additionally, Co−B/Ru₁₂ still maintains the highest FE_NH₃ compared to catalysts with higher Ru content, such as Co−B/Ru₃₈ and Co−B/Ru₇₅ (Fig. S15a, SI).

The NH₃ yield rates ($Y_{NH_3}$) of Co−B, Co−B/Ru₁₂, and Ru are presented in Fig. 2d. Notably, Co−B/Ru₁₂ stands out among Co−B and Ru, showing a significantly higher $Y_{NH_3}$ of 15.0 ± 0.7 mg h⁻¹ cm⁻² which is ≈15.2-fold that of Co−B and ≈6.0-fold that of Ru (−0.2 V). Meanwhile, the partial current density for NH₃ production ($J_{NH_3}$) of Co−B/Ru₁₂ easily reached over −100 mA cm⁻², a crucial metric for assessing the

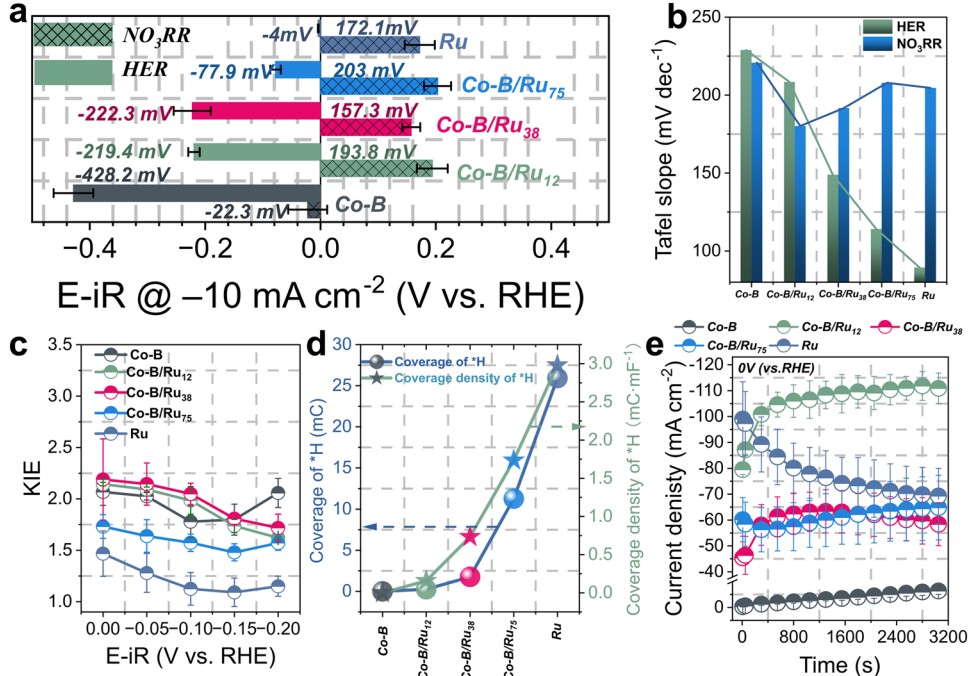

**Fig. 3 | Evaluation of the NO₃RR reaction kinetics and modulation of *H on Co–B/Ru₁₂. a** LSV-derived potentials at a current density of −10 mA cm⁻² for different catalysts for both HER and NO₃RR. The uncompensated resistance, determined by potentiostatic electrochemical impedance spectroscopy, was 18.97 ± 7.69 ohm in 0.1 mol L⁻¹ NaOH, 10.53 ± 2.32 ohm in 0.1 mol L⁻¹ NaOH + 0.1 mol L⁻¹ NaNO₃. **b** Tafel slopes derived from LSVs for both HER and NO₃RR. **c** KIE in the potential window from 0 to −0.2 V (vs. RHE). **d** *H coverage and corresponding coverage density normalized by the $C_{dl}$ for different catalysts. **e** Chronoamperometry measurements at a potential of 0 V (vs. RHE) in 0.1 mol L⁻¹ NO₃⁻ and 0.1 mol L⁻¹ NaOH. Error bars denote the standard deviations from at least three independent measurements. Source data for this figure are provided as a Source Data file.

highly efficient conversion of NO₃RR[24,34], starting from a low potential of −0.05 V. In contrast, both Co–B and Ru failed to achieve this metric within the applied potential range (Fig. S15b, SI). Additionally, it exhibits a linear potential-dependent increase in $Y_{NH3}$, contrasting with the observed trend for Ru. This difference hints that Co−B/Ru₁₂ prefers NO₃RR rather than competitive reactions such as HER as the overpotential is increasing. The Co−B/Ru₃₈ and Co−B/Ru₇₅ bimetallic samples also exhibit different levels of potential-dependent increase in $Y_{NH3}$ and $J_{NH3}$ (Fig. S15c, d, SI). These results further highlight the synergistic effect between Co−B and Ru in the bimetallic catalyst, which can greatly enhance and accelerate the conversion of NO₃⁻ to NH₃ and suppress the HER within the applied potential range.

We achieved a peak NH₃ half-cell EE of 40.9 ± 4 % (theoretical value being 43.9%) on Co−B/Ru₁₂ at 0 V, showing a 1.4-fold and 9.9-fold improvement compared to the case of the pure Ru and Co−B, respectively (Fig. 2e). Furthermore, Co−B/Ru₁₂ maintains its high half-cell EE across the applied potential range. This superior EE, together with the FE$_{NH3}$ and $Y_{NH3}$ in a low overpotential range, endow the proposed Co−B/Ru₁₂ as one of the top-ranking NO₃RR electrocatalysts for synthesizing NH₃ (Table S1, SI), even when compared other NH₃ production pathways (Table S2, SI). Furthermore, the high performance of Co−B/Ru₁₂ remained consistent during chronoamperometry at a potential of −0.1 V. As shown in Fig. S16 (SI), the catalyst exhibits a relatively stable trend in preserving its $Y_{NH3}$ with high EE after exchanging the electrolyte every 2 h.

To pursue the higher performance of NO₃RR, the impact of various initial nitrate concentrations (CNO₃⁻) on the FE$_{NH3}$ and $Y_{NH3}$ at Co−B/Ru₁₂ was further investigated. The catalyst shows negligible changes in the FE$_{NH3}$ when CNO₃⁻ increases from 0.1 to 0.2 mol L⁻¹ within the potential window from 0 to −0.10 V (Fig. 2f). However, the FE$_{NH3}$ begins to decrease at higher CNO₃⁻ levels of 0.5 mol L⁻¹ and 1.0 mol L⁻¹. In terms of $Y_{NH3}$, a volcano-type trend is observed with varying CNO₃⁻ (Fig. 2g). The catalyst achieves its highest $Y_{NH3}$ of

47.6 ± 2.7 mg h⁻¹ cm⁻¹ and a high $J_{NH3}$ of 600.0 ± 34.0 mA cm⁻² (Fig. S17, SI) at −0.1 V (vs. RHE) in 0.5 mol L⁻¹. These results indicate that the conversion selectivity and yield do not follow a simple linear correlation with the reactant concentration. Notably, the catalyst still retains a $Y_{NH3}$ of 46.1 mg h⁻¹ cm⁻¹ and a FE$_{NH3}$ of 85.1% after 15 cycles stability test and maintains a $J_{NH3}$ of around 600 mA cm⁻² throughout the 30 h measurements (Figs. 2h and S18, SI).

To determine whether the source where the NH₃ is derived from NO₃⁻ or from impurities present in the electrolyte or air, we employed ¹H NMR to investigate the NH₃ formation on Co−B/Ru₁₂ in more detail (Fig. S19, SI). The ¹H NMR spectrum revealed a distinct doublet of ¹⁵NH₄⁺ when using ¹⁵NO₃⁻, conclusively confirming that the generated ¹⁵NH₄⁺ indeed originated from the reduction of NO₃⁻ (Fig. S19a, SI). The quantity of ¹⁴NH₄⁺ measured by ¹H NMR closely aligns with the amount determined by UV−Vis spectrophotometry, thereby validating the reliability of the results (Fig. S19d, SI).

## Understanding high-rate NH₃ generation on Co−B/Ru₁₂

To rationalize the superior FE$_{NH3}$ and $Y_{NH3}$ over Co−B/Ru₁₂ in a low overpotential range, we compared the overpotentials of the investigated five catalysts required to achieve a current density of 10 mA cm⁻² in the kinetic area with negligible mass transport limitation for both HER and NO₃RR assuming that HER is the most challenging competitive reaction (Fig. S20, SI).

Co−B shows a negative potential of −22.3 mV for catalyzing NO₃RR, while the catalysts containing Ru show a notably more positive potential (e.g.,193.8 mV for Co−B/Ru₁₂) (Fig. 3a), confirming that Ru plays a significant role in enhancing the kinetics of the NO₃RR. Pure Ru exhibits the lowest overpotential for the HER and as the ratio of Co−B within the bimetallic catalysts increases, the HER overpotentials required to reach 10 mA cm⁻² also increase. This implies that the presence of Co−B impedes a high reaction kinetics for the HER. The Tafel slopes (Figs. 3b and S21, SI) show a linear trend. With the incremental

increase of the Co content, the HER Tafel slopes show a gradual increase from 89.4 mV dec$^{-1}$ for Ru to 228.9 mV dec$^{-1}$ for Co–B, suggesting that Co–B inhibits the HER kinetics, which is further supported by in-situ differential electrochemical mass spectrometry (DEMS) analysis, revealing a notably higher negative potential for H$_2$ generation for catalysts containing Co–B (Fig. S22, SI). Simultaneously, Co–B/Ru$_{12}$ exhibits the lowest Tafel slope for NO$_3$RR. Due to NO$_3$RR being a series of proton-coupled electron transfer reactions, we hypothesize that the superior reaction kinetics for Co–B/Ru$_{12}$ can be ascribed to the synergistic effect between Co–B and Ru. This synergy likely enhances the generation of an optimal quantity of adsorbed hydrogen (*H) on the catalyst surface, thereby promoting the reaction kinetics of the hydrogenation process during the NO$_3$RR while concurrently inhibiting the HER.

Kinetic isotope experiments (KIE) were further derived using deuterated water (D$_2$O) and deuterated hydroxide (OD$^-$) to reveal both *H generation and the transfer rate of the studied catalysts during the NO$_3$RR. The KIE determines whether the generated *H from water dissociation can be consumed by N-containing intermediates or is undergoing recombination to form hydrogen (Fig. S23, SI). The LSV-derived KIE values (Fig. 3c) are exceeding one for all studied catalysts, implying that proton transfer rather than electron transfer is involved in the rate-determining step during NO$_3$RR. Catalysts containing Co–B exhibit significantly higher KIE values compared to Ru within the potential range of 0 to −0.2 V (vs. RHE), implying that the Co–B based catalysts may encounter a larger barrier during the generation of *H from water dissociation or in the subsequent transfer of the produced *H during the hydrogenation step during the NO$_3$RR. We investigated the *H coverage on the catalysts by integrating the *H desorption peak area observed in cyclic voltammograms (Fig. S24, SI). The *H desorption peak was observed primarily for the catalyst containing Ru (inset of Fig. S24, SI), suggesting that the produced *H may be largely associated with Ru[35]. In Fig. 3d, pure Ru exhibits the highest *H coverage of 25.9 mC, which decreases sharply with increasing Co–B content. This decrease remains significant even when normalized by the double-layer capacitances ($C_{dl}$) of the catalysts (Fig. S25, SI). These findings indicate that pure Ru has the fastest *H generation rate from water dissociation, while the presence of Co–B in the bimetallic catalyst can effectively modulate the *H generation rate to avoid the recombination of two *H atoms to form H$_2$. Furthermore, as shown in Fig. S26 (SI), the Co–B/Ru$_x$ catalyst exhibits higher NH$_3$ partial currents with KIE values located between those of Co–B and pure Ru. This suggests an optimized proton transfer rate resulting from the interaction between Co–B and Ru, which favors the requirement of the hydrogenation step for NO$_3$RR rather than HER[62]. To investigate the possible active site for adsorbing NO$_3^-$, we compared the $J_{NH3}$ normalized by the $C_{dl}$ for Co–B and Ru (Fig. S27a, SI), indicating that both Co–B and Ru have varying levels to adsorb NO$_3^-$ and convert it to NH$_3$ at different potentials. Fig. S27b (SI) demonstrates that the $J_{NH3}$ of Co–B/Ru$_{12}$ is much higher than that of the individual components, indicating that the effect cannot be simply explained by the individual contribution of Co–B and Ru. Therefore, attributing the active sites responsible for adsorbing NO$_3^-$ and converting it to NH$_3$ solely to either Ru or Co–B may be too simple.

The significantly enhanced FE$_{NH3}$ observed for Co–B/Ru$_{12}$ can be attributed to the introduction of Co–B, which effectively suppresses the competitive HER by adequately generating *H through water dissociation. This provision of *H precisely enables the hydrogenation step for the NO$_3$RR. Simultaneously, the synergetic effect present in Co–B/Ru$_{12}$ boosts the reaction kinetics for the NO$_3$RR. On the other hand, to comprehend the high $Y_{NH3}$ observed for Co–B/Ru$_{12}$, we compared the chronoamperometry results of the investigated catalysts during NO$_3$RR at 0 V vs. RHE, a sufficiently high potential to avoid the influence of the HER. All catalysts containing Co–B exhibit a distinct activation process. Whereas Ru is undergoing a deactivation

process (Fig. 3e) possibly due to the excessive formation of *H on the Ru surface, which cannot be consumed by the hydrogenation step of NO$_3$RR, covering the active sites and hindering the adsorption and conversion of NO$_3^-$ [2,63]. Specifically, for Co–B/Ru$_{12}$, the current density increased by 1.4-fold, reaching −110.9 ± 5.9 mA cm$^{-2}$ after 3000 s, while for Ru the current density decreased by 0.7-fold (Fig. S28a, SI). The current density for Co–B increased by 22-fold but remained at a low current density of −6.4 ± 1.4 mA cm$^{-2}$ (Fig. S28b, SI).

## Single-entity electrochemistry

Understanding changes in the catalyst's intrinsic activity and further establishing the inherent structure-electrochemical property relationship of the catalyst is crucial for clarifying its key role in catalytic mechanisms. The primary challenge in determining the intrinsic activity of the electrocatalyst lies in maintaining a stable local chemical environment, e.g. concerning the local pH value at the catalyst-electrolyte interface throughout the reaction. This aspect is particularly crucial for proton-coupled electron transfer reactions such as the NO$_3$RR. However, measuring catalytic activity using macroscopic electrodes does not fully address this challenge due to the slow mass transfer process caused by planar diffusion (Fig. 4a). Nanoelectrodes are an ideal platform to directly investigate the intrinsic activity of single catalyst entities because changes in the local chemical environment during the reaction are prevented due to the fast hemispherical diffusional mass transfer[64,65]. Combination with IL-TEM allows for the direct investigation of structural changes induced by the electrocatalytic reaction and facilitates to deriving the relationship between the structure and electrochemical properties of the catalyst. We utilized a previously developed single-nanoparticle-on-a-nanoelectrode technique to place a single entity of Co–B/Ru$_{12}$, on top of a carbon nanoelectrode (CNE), thus enabling the investigation of its intrinsic activity changes and corresponding structural changes during NO$_3$RR (Fig. 4b)[55,56,58,66]. Specifically, a micromanipulator tip controlled by a controller was used to select and pick individual Co–B/Ru$_{12}$ particles from the sample stage, then precisely positioning them onto a FIB-processed CNE under SEM control (Figs. S29–S31, SI). TEM images (Fig. S32, SI) show the successful fabrication of two CNE@Co–B/Ru$_{12}$ assemblies. The intrinsic NO$_3$RR activity of the CNE@Co–B/Ru$_{12}$ assemblies was investigated by cyclic voltammetry (CV) in 0.1 mol L$^{-1}$ NaOH containing 0.1 mol L$^{-1}$ NaNO$_3$. A maximum of 10 CV cycles was applied to a single particle due to the much faster reaction rate and the speed of structural evolution compared to macroelectrodes, as well as to decrease the likelihood of single particles detaching from the CNE[55,57]. The insets of Fig. 4c, d show the 1st CVs of the two CNE@Co–B/Ru$_{12}$ nano-assemblies exhibiting a significantly higher NO$_3$RR current and lower overpotential compared to a bare CNE, thus excluding the possible interference from the bare CNE.

For both CNE@Co–B/Ru$_{12}$ nano-assemblies a distinct plateau current at around 0 V (vs. RHE) is observed starting from the 2nd CV scan (Fig. 4c, d). The ultra-fast mass transfer toward the nanoelectrode can rule out that the plateau is caused by diffusion limitations. Instead, it is due to the maximum turnover of the fully occupied active sites on the Co–B/Ru$_{12}$ particle, which is widely used to precisely estimate the electrochemically active size for a given reaction on nanoelectrodes[67–69]. Since the plateau potential of around 0 V (vs. RHE) is not allowing HER, the plateau current can serve as an indicator of the number of active sites of the Co–B/Ru$_{12}$ single entity for the NO$_3$RR. The plateau current shows a continuous increase with successive CV cycles, indicating a progressive increase in the number of active sites for the NO$_3$RR, which perfectly aligns with the findings discussed in Fig. 3e. Notably, the increasing plateau observed on the nano-electrode is not present on the macro-electrode under the same scan rate (Fig. S33, SI), highlighting the technique's unique advantage in determining catalyst's intrinsic changes. Furthermore, the ratio of the change in the reduction plateau current with respect

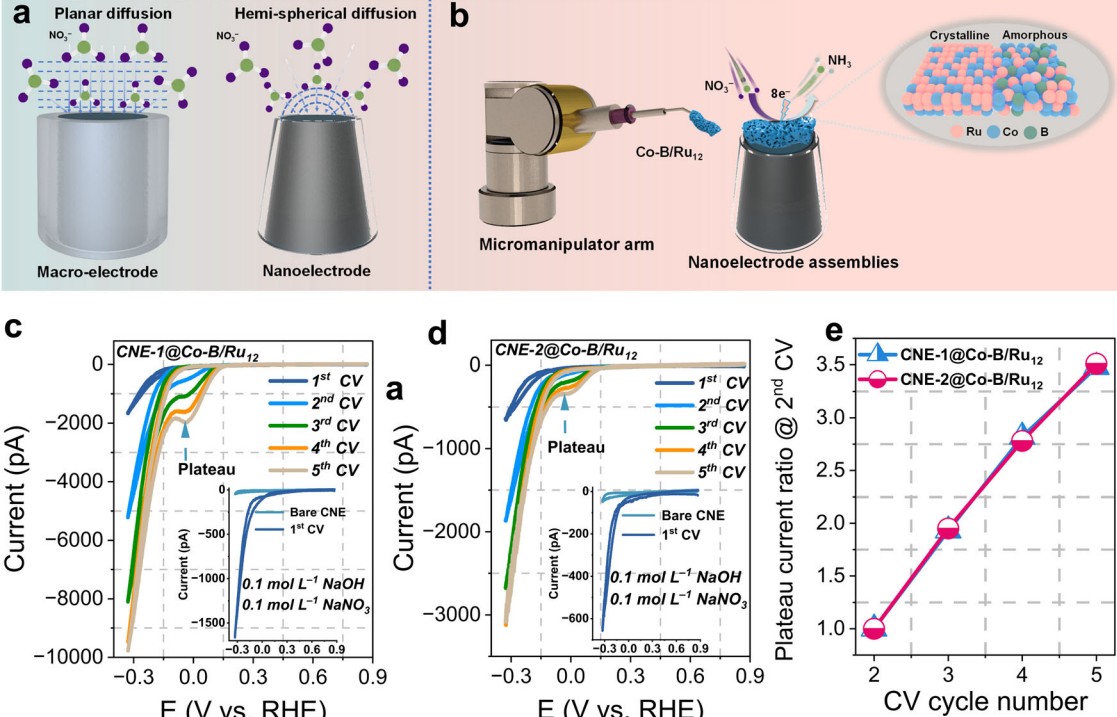

**Fig. 4 | Single-entity electrochemistry of Co–B/Ru$_{12}$. a** Scheme of diffusional flux toward a macroelectrode and a nanoelectrode. **b** Illustration of the fabrication of Co−B/Ru$_{12}$ CNE nanoelectrode assemblies. **c** CVs of CNE-1@Co-B/Ru$_{12}$ in 0.1 mol L$^{-1}$ NaOH containing 0.1 mol L$^{-1}$ NaNO$_3$ at a scan rate of 50 mV s$^{-1}$. **d** CVs of CNE-2@Co−B/Ru$_{12}$ in 0.1 mol L$^{-1}$ NaOH containing 0.1 mol L$^{-1}$ NaNO$_3$ at a scan rate of 50 mV s$^{-1}$. The insets in **c** and **d** show the 1$^{st}$ CVs of the bare CNE and the corresponding nanoelectrode assemblies. **e** Plateau current ratio normalized by the 2$^{nd}$ CV scans of CNE-1@Co-B/Ru$_{12}$ and CNE-2@Co-B/Ru$_{12}$. The potentials in (**c**) and (**d**) are not iR-corrected due to the current being in the pA range. Source data for this figure are provided as a Source Data file.

to that from the corresponding 2$^{nd}$ CV scan is a measure of the number of active sites for NO$_3$RR with an increasing number of CV cycles. As shown in Fig. 4e, a linear and size-independent increase in the number of active sites for the NO$_3$RR could be derived with successive catalyst activation.

IL-TEM was used to investigate the reasons for the increase in active sites by comparing the structure of the same CNE@Co–B/Ru$_{12}$ particle before and after five CV cycles. Minimal alterations in the overall shape of a single particle were observed before and after NO$_3$RR (Fig. 5a, b). However, closer examination at higher magnification disclosed a structural reconstruction (see labeled regions in Fig. 5c). This is particularly noticeable in the region marked by the green rectangle, where a partial structural collapse after five CV cycles can be noticed. The EDS mapping within this rectangular region indicates a redistribution of elements, which is particularly noticeable for Co (Fig. S34, SI). These findings suggest a correlation between the structural reconstruction and the corresponding continuous activation process during the NO$_3$RR. To better comprehend this correlation, we subjected the CNE-2@Co−B/Ru$_{12}$ nano-assembly to ten CV cycles and monitored its structural changes at each five CVs interval using IL-TEM and EDS mapping. CNE-2@Co−B/Ru$_{12}$ shows a consistent increase in the plateau current from the 6$^{th}$ to the 10$^{th}$ CV cycles (Fig. S35a, SI), indicating a continuous further exposure of active sites for the NO$_3$RR. As shown in Fig. S35b (SI), the ratio of the plateau current change ratio exhibits a significantly steeper increase during 6$^{th}$ to 8$^{th}$ CV cycles, followed by a slower increase in the last two CV cycles, suggesting that the activation and the increase in the accessible number of active sites approaches a limit.

The structural changes of CNE-2@Co−B/Ru$_{12}$ exhibit a different trend during the two electrochemical stages as shown in Fig. 5d−f and Fig. S36 (SI). At the later stage (6$^{th}$ CV to 10$^{th}$ CV), a more pronounced structural change becomes visible, with the structure becoming

looser, particularly in the region highlighted by the arrow. This change further provides visible evidence supporting the exposure of more electrochemical surface area. EDX mapping unveils a progressive Co leaching, while Ru shows minor changes (Figs. 5g–l and S37, SI). This suggests that Co leaching could be responsible for inducing the structural reconstruction of the catalyst, which is further evidenced by the much higher concentration of Co compared to Ru in the electrolyte after 1 h of electrolysis at different potentials (Fig. S38a, b, SI). However, B shows a high concentration in the electrolyte during the initial 1 h of electrolysis at 0 V (vs. RHE), while maintaining a level similar to the blank electrolyte during subsequent electrolysis at −0.05 V and −0.10 V (vs. RHE). This indicates that B leaches into the electrolyte and does not redeposit onto the catalyst layer to continue contributing to the leaching process as Co does (Fig. S38c, SI). Therefore, the impact of B as a continuous active site in our catalyst is negligible. The leached cobalt ions can redeposit onto the catalyst or electrode surface from the electrolyte, forming a reconstructed structure (Fig. S39, SI). TEM images of the catalyst on carbon paper after 10 h stability measurement show that the reconstructed structure forms a nanosheet matrix composed of Co(OH)$_2$ with a coverage of Ru (Fig. S40, SI). A similar structure was observed after extended stability tests (Fig. S41, SI).

## Discussion

In summary, our study suggests a high-performance Co-B/Ru$_{12}$ electrocatalyst for the NO$_3$RR, which displays an excellent FE$_{NH3}$ while operating at a low overpotential. Its superior performance can be ascribed to the optimized NO$_3$RR reaction kinetics facilitated by the Ru sites and modulated coverage of *H due to the role of Co-B in inhibiting water dissociation, thus suppressing the undesired competing HER. The activation process occurring during NO$_3$RR significantly contributes to the high NH$_3$ yield rate at Co−B/Ru$_{12}$. Utilizing SEE

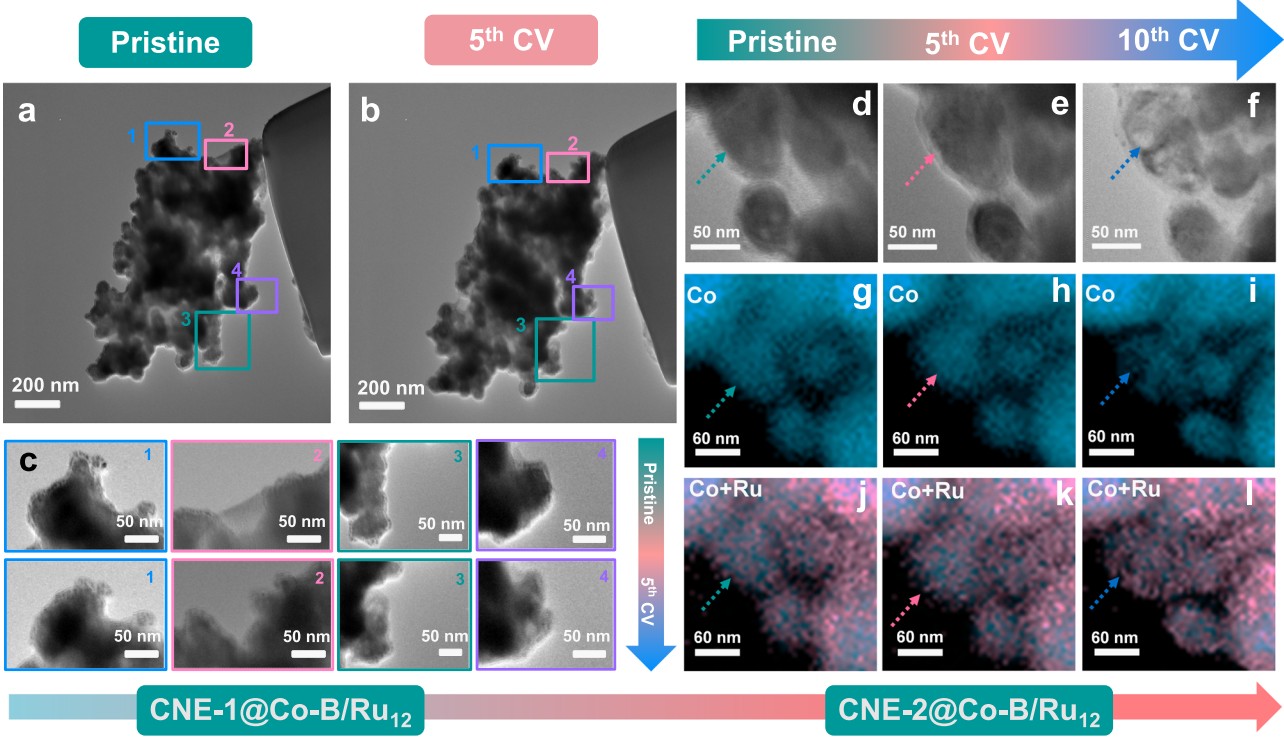

**Fig. 5 | Structural reconstruction of Co−B/Ru$_{12}$ single particles.** TEM images of CNE-1@Co−B/Ru$_{12}$ before (**a**) and after (**b**) five CV cycles in 0.1 mol L$^{-1}$ NaOH containing 0.1 mol L$^{-1}$ NaNO$_3$ at a scan rate of 50 mV s$^{-1}$. **c** Zoom in on the indicated regions in (**a**) and (**b**); the identical regions in (**a**) and (**b**) were labeled with the same number and color for easier comparison. TEM images of CNE-1@Co−B/Ru$_{12}$ before (**d**), after the 5$^{th}$ CV cycle (**e**), and after the 10$^{th}$ CV cycles (**f**) in 0.1 mol L$^{-1}$ NaOH containing 0.1 mol L$^{-1}$ NaNO$_3$ at a scan rate of 50 mV s$^{-1}$. EDX mapping of Co of CNE-1@Co−B/Ru$_{12}$ before (**g**), after the 5$^{th}$ CV cycle (**h**), and after the 10$^{th}$ CV cycle (**i**) in 0.1 mol L$^{-1}$ NaOH containing 0.1 mol L$^{-1}$ NaNO$_3$ at a scan rate of 50 mV s$^{-1}$. Overlay EDX mapping of Co and Ru for CNE-1@Co−B/Ru$_{12}$ before (**j**), after the 5$^{th}$ CV cycle (**k**), and after the 10$^{th}$ CV cycle (**l**) in 0.1 mol L$^{-1}$ NaOH containing 0.1 mol L$^{-1}$ NaNO$_3$ at a scan rate of 50 mV s$^{-1}$. Source data for this figure are provided as a Source Data file.

coupled with IL-TEM confirms the activation process. The reason for the observed activation originates from continuously exposing more active sites induced by in-situ electrochemical reconstruction likely resulting from Co leaching during the NO$_3$RR. We believe that the detailed insights into the in-situ electrochemical reconstruction and the modulation of *H coverage on Co−B/Ru$_{12}$ shed light on possibly more rational catalyst design strategies not only for the NO$_3$RR but also for other energy conversion reactions.

## Methods
### Chemicals and materials
Maleic acid (≥99%) was purchased from Riedel-de Haën. Potassium sulfate (≥98%) and hypochlorite solution (14% Cl$_2$ in aqueous solution) were from VWR Prolabo Chemicals. Sodium citrate dihydrate (≥99%) and phosphoric acid (85wt%) were from J.T. Baker. Sodium borohydride (≥98%) and salicylic acid (≥99%) were from Merck. Sodium hydroxide (≥98%), sodium nitrate (≥98%), and sodium nitrite (≥99.0%) were obtained from Carl Roth. Ruthenium(III) chloride hydrate (≥99.98%), cobalt(II) chloride (≥98.0%), Nafion perfluorinated resin solution (5wt%), ammonium chloride (≥99.998%), ammonium-$^{15}$N chloride (99 atom%; ≥98%), sodium nitrate-$^{15}$N (99 atom%; ≥98%), sodium nitroferricyanide, N-(1-naphthyl) ethylenediamine dihydrochloride (≥98%), sulfonamide (≥99%), deuterium oxide (99.9 atom% of D), sodium deuteroxide solution (40wt% 99.5% of D), dimethyl sulfoxide-d6 (99.9 atom %D), hydrochloric acid (≥37%), sulfuric acid (≥99.999%) were from Sigma-Aldrich. Carbon paper (H23-C9) was from Freudenberg Performance Materials. The quartz capillaries with an outer diameter of 1.2 mm and an inner diameter of 0.9 mm with a total length of 7.5 cm were from Science Products. All chemicals were used as received without further purification.

### Synthesis of Co−B/Ru$_{12}$
Twenty milliliters of a mixed solution of CoCl$_2$ (0.08 mol L$^{-1}$) and RuCl$_3$·xH$_2$O (0.02 mol L$^{-1}$) was flushed with argon and maintained at 0 °C using an ice-bath in a round-bottomed Schlenk flask. A 20 ml 0.3 mol L$^{-1}$ NaBH$_4$ in 0.1 mol L$^{-1}$ NaOH solution was added dropwise into the mixed solution. The formation of a dark precipitate was observed instantaneously. After 30 min, the precipitate was collected by centrifugation and washed several times with deionized water and ethanol before it was dried at 70 °C. The Co−B/Ru$_{38}$ and Co−B@Ru$_{75}$ samples were prepared by varying the molar ratios in the precursor solutions. Specifically, for Co−B@Ru$_{38}$, a molar ratio using CoCl$_2$ (0.05 mol L$^{-1}$) to RuCl$_3$·xH$_2$O (0.05 mol L$^{-1}$) was utilized, while for Co−B/Ru$_{75}$, the molar ratio was adjusted to CoCl$_2$ (0.02 mol L$^{-1}$) and RuCl$_3$·xH$_2$O (0.08 mol L$^{-1}$). Additionally, Co−B and Ru were synthesized using a similar method, involving only CoCl$_2$ for Co−B and only RuCl$_3$·xH$_2$O for Ru in their respective precursor solutions.

### Material characterization
X-ray diffraction (XRD) data were obtained using a Bruker D8 Discover X-ray diffractometer with a Cu Kα radiation source ($\lambda$ = 1.5418 Å) in the range 2θ = 10–90° and a step size of 0.02°. Data evaluation has been performed with the DIFFRAC.EVA software using the PDF-2 database. Scanning electron microscopy (SEM) images were obtained using a Quanta 3D FEG scanning electron microscope at 30.0 kV in the high-vacuum mode. The SEM is equipped with a focused ion beam (FIB) and micromanipulator for enhanced functionality. TEM images, scanning TEM images, and energy dispersive X-ray (EDX) elemental mapping were conducted in a JEOL microscope (JEM-2800) with a Schottky-type emission source at 200 kV. Dual SDD X-ray detectors are used to capture EDS signals for elemental mapping with a solid angle of 0.95 srad, and with 133 eV of spectral resolution. For EELS measurement,

the probe-corrected JEOL JEM-ARM200F NEOARM system was employed, featuring a cold field-emission gun (FEG) with an acceleration voltage ranging from 80 to 200 kV. The EELS resolution of the Gatan Quantum system is 0.3 eV. The background and noise subtraction and elemental signal was processed with Gatan DigitalMicrograph GMS 3.60. The elemental content was investigated by inductively coupled mass spectrometry (ICP-MS). Nuclear magnetic resonance (NMR) spectroscopy was conducted on a Bruker 400 MHz NMR spectrometer. XPS was carried out with an AXIS Nova spectrometer (Kratos Analytical) equipped with a monochromatic Al Kα X-ray source (1487 eV, 15 mA emission current), maintaining a chamber pressure of $\approx 2 \times 10^{-9}$ mbar. The high-resolution spectra of the core levels were collected in the fixed transmission mode at a pass energy of 20 eV. Charging effects were compensated using a flood gun. The C 1s peak of adventitious carbon at 284.8 eV was used for calibration of the binding energies.

## Carbon nanoelectrodes (CNE)

The quartz capillaries were pulled using a P-2000 laser puller (Sutter Instruments) with the parameters: Heat = 740, Filament = 4, Velocity = 45, Delay = 130, and Pull = 10. Subsequently, the CNEs were fabricated by depositing carbon inside the quartz capillaries by pyrolysis of a gas mixture of propane (technical grade) and n-butane (99.5%) under argon counter-flow (99.999%, 50 mL min$^{-1}$). Finally, these electrodes were further processed by FIB milling in the SEM to obtain a controlled flat carbon disk-shaped nanoelectrode.

## Single particle nanoelectrode assemblies (CNE@Co−B/Ru$_{12}$)

Single particle nanoelectrode assemblies were fabricated using a micromanipulator within an SEM chamber[55,56,58]. The particles were uniformly dispersed on an ultra-flat gold-coated Si-wafer. The micromanipulator, equipped with a robotic arm capable of precise three-dimensional movement along the x, y, and z axes, allows for fine control. The sample is mounted flat on the SEM stage, with the CNEs vertically aligned to the tip, ensuring that the disk-shaped surface of the CNE is collinear with it (Fig. S30, SI). A controller operates the micromanipulator arm inside the SEM, enabling the selection and precise placement of a single particle from the wafer surface onto the targeted position on the tip of a FIB-cut CNE.

## Electrochemical NO$_3$RR measurements

Electrochemical tests on a macroelectrode (carbon paper) were conducted in a three-electrode configuration using an H-type cell and an Autolab PGSTAT204 potentiostat/galvanostat (Metrohm). The data obtained from the Autolab PGSTAT204 were saved as.txt files and subsequently plotted using Origin (OriginLab). The electrolyte was prepared by dissolving 0.1 molar sodium hydroxide with varying concentrations of sodium nitrate ranging from 0.1 to 1 molar in a 1 L volumetric flask. The solution was then stored in a clean plastic bottle at room temperature overnight before being used for electrochemical measurements. The anode compartment (15 mL of electrolyte) and the cathode compartment (30 mL of electrolyte) in an H-type cell were separated by Nafion®117 membrane (DuPont, thickness: 0.178 mm, size: 1.5 × 1.5 cm²). Before testing, the Nafion®117 membrane was pretreated in 5% H$_2$O$_2$ solution at 80 °C for 1 h, then in 0.5 M H$_2$SO$_4$ solution at 80 °C for another 1 h, and finally washed with deionized water several times. Catalyst-modified carbon paper with a geometric area of 0.25 cm$^{-2}$ was used as the working electrode (WE). Catalyst inks were prepared by dispersing the catalyst powder (10 mg) in a mixture of absolute ethanol (980 μl) and 5% Nafion (20 μl), followed by ultrasonication for 20 min. A 12.5 μl ink was dropped onto the cleaned carbon paper to form a catalyst layer with a mass loading of 0.5 mg cm$^{-2}$. The carbon paper surface was covered with a dense layer of carbon powder mixed with PTFE, creating a relatively flat and dense surface forming a homogenous catalyst layer. A double-junction Ag/

AgCl/3 M KCl electrode was used as the reference electrode. The potential of this Ag/AgCl/3 M KCl electrode was measured against a well-maintained commercial Ag/AgCl/3 M KCl reference electrode before each measurement to ensure proper function. A Pt mesh was used as the counter electrode. Measured potentials were compensated by the 80% $iR_u$-drop and converted to the RHE with according to the equation: $E_{RHE} = E_{Ag/AgCl/3MKCl} + 0.207 V + 0.059 \ast pH - iR_u$. $E_{RHE}$ is the potential vs. RHE, $E_{Ag/AgCl/3MKCl}$ is the potential vs. the reference electrode Ag/AgCl/3 M KCl, $R_u$ is the uncompensated resistance, $i$ is the current and 0.207 V is the standard potential of the reference electrode. The uncompensated solution resistance $R_u$ was determined by means of electrochemical impedance spectroscopy with an ac perturbation of 10 mV$_{PP}$ in the frequency range from 50 kHz to 1 Hz. The values obtained were 18.97 ± 7.69 ohm in 0.1 M NaOH, 10.53 ± 2.32 ohm in 0.1 M NaOH + 0.1 M NaNO$_3$, 4.78 ± 0.25 ohm in 0.1 M NaOH + 0.2 M NaNO$_3$, 4.43 ± 0.36 ohm in 0.1 M NaOH + 0.5 M NaNO$_3$, 4.39 ± 0.16 ohm in 0.1 M NaOH + 1.0 M NaNO$_3$. The pH value was calculated based on the concentration of OH$^-$ in the electrolyte containing 0.1 M NaOH. Using the equation pH = 14−pOH and pOH = −log(0.1), the pOH is 1, resulting in a pH of 13. Linear sweep voltammograms were recorded at a scan rate of 5 mV s$^{-1}$. The current was normalized to the geometric area (0.25 cm$^{-2}$). Chronoamperometry tests were performed for 1 h in 30 mL catholyte electrolyte at a stirring rate of 300 rpm. The resulting electrolyte was collected and stored at 4 °C before analysis. The long-term performance of the Co−B/Ru$_{12}$ catalysts was evaluated by 15 cycles of 2 h chronoamperometry at −0.1 V (vs. RHE) in 0.5 M NO$_3^-$, the electrolyte was collected for analysis after each 2 h electrolysis cycle. After cleaning the cathodic compartment three times with deionized water, fresh electrolyte was added to the catholyte for the next cycle.

Single-particle electrochemical measurements on a CNE were performed inside a Faraday cage using a Modulab potentiostat (Solartron Analytical) in a two-electrode configuration, with a self-made double-junction Ag/AgCl/3 M KCl electrode serving as both the reference and counter electrode. The data from the Solartron Analytical were saved as.csv files and then plotted using Origin (OriginLab).

## Differential electrochemical mass spectrometry (DEMS) measurements

DEMS measurements were performed using a custom-made two-compartment 3D-printed cell with the WE integrated into the base of the cell and its surface being approached by a custom-made mass-spectrometry (MS) tip assembly consisting of a 5 mm silica frit with ~160 μm pore size embedded in a glass tube and covered with a hydrophobic PTFE membrane with a pore size of 20 μm and a thickness of 50 μm (Cobetter, Cat. No. PF-002HS) (Fig. S42, SI). This MS tip assembly worked as the interface between the electrolyte and the vacuum system of the MS. The distance between the tip and the WE was controlled using an optical microscope monochrome camera (The ImagingSource). The H$_2$ signal with a mass-to-charge ratio ($m/z = 2$) was recorded using a GAM 400 (InProcess Instruments) mass spectrometer with an SEM voltage of 1400 V and an emission current of 1 mA. LSV with a scan rate of 2 mV s$^{-1}$ and a step size of 5 mV was conducted at the WE. The hydrogen detection potential was determined when the signal was three times the noise.

## CV measurements for *H coverage

The coverage of *H was measured by CV with a scan rate of 20 mV s$^{-1}$ in 0.1 mol L$^{-1}$ NaOH. A stable CV was obtained after around ten cycles. To quantify the amount of *H on the catalyst surface, it was assumed that all *H atoms are completely desorbed from the electrode surface during the CV. Hence, the total charge of the desorption peak of *H can be calculated by integrating $I$ vs. $t$. The potential axis in Fig. S24 can be converted to a time axis by the equation of $t = (E(t) − E(0))/v$, where $E(t)$ refers to the potential (V) at time $t$ (s), $E(0)$ is the starting potential, $v$ is the scan rate.

## Determination of the double-layer capacitance

Cyclic voltammograms were recorded at various scan rates (60, 90, 120, 150, 180, 210 mV s$^{-1}$) to derive the double-layer capacitance ($C_{dl}$) by plotting $\Delta J = (J_{anodic} - J_{cathodic})/2$ against the scan rate, where $J_{anodic}$ and $J_{cathodic}$ are the anodic and cathodic current densities, respectively.

## Detection of ammonia

NH$_3$ was quantified using the indophenol blue method with slight modification. First, 0.2 mL of the collected electrolyte sample after electrolysis was diluted to 2 mL. Then, 2 mL of 1 mol L$^{-1}$ NaOH, sodium citrate, and salicylic acid were mixed, and 1 mL freshly prepared 0.05 mol L$^{-1}$ NaClO was added. The solution was shaken for a few seconds and 0.2 mL of a 1 wt.% sodium nitroferricyanide was added and left to rest for 1 h at room temperature. Finally, the absorption of the mixed solution was measured in a UV–Vis spectrophotometer at 655 nm. A calibration curve was made by using a series of standard NH$_4$Cl (99.998%) solutions.

## Detection of nitrite

The nitrite concentration was determined by UV–Vis spectrophotometry. First, 1 mL 1 mol L$^{-1}$ HCl was added to 5 mL of diluted post-electrolysis electrolyte sample, and then 0.1 mL of the color reagent (0.20 g N-(1-naphthyl)ethylenediamine dihydrochloride (≥98%), 4.0 g of sulfonamide (≥99%), and 10 mL of phosphoric acid (85wt.% in H$_2$O) in 50 mL of deionized water) were added. After 20 min at room temperature the absorption at 540 nm was detected by means of UV–Vis spectrophotometry. For NO$_2^-$ quantification a calibration curve was obtained using a series of standard sodium nitrite (≥96%) solutions.

## Calculation of the FE, $Y_{NH3}$, $J_{NH3}$

The FEs of the NO$_3$RR toward NH$_3$ and NO$_2^-$ are defined as the charge for the formation of NH$_3$ and NO$_2^-$ divided by the total charge passing through the WEs during electrolysis. The FE and yield rate of NH$_3$ were calculated as follows: $FE_{NH3} = (8*F*C_{NH3}*V)/Q$, $Y_{NH3} = (C_{NH3}*V)/(A*t)$, $J_{NH3} = (Q*FE_{NH3})/(A*t)$, where $F$ is the Faraday constant (96485.3 C mol$^{-1}$), $C_{NH3}$ represent the concentration of the detected NH$_3$ (mol L$^{-1}$), $A$ is the geometric electrode area (cm$^2$), $V$ is the volume of the catholyte (L), and $t$ is the reaction time (h). The FE of NO$_2^-$ was calculated as follows: $FE_{NO2-} = (2*F*C_{NO2-}*V)/Q$ since two electrons are involved in the reduction for NO$_2^-$.

## Calculation of the cathodic energy efficiency

The half-cell EE is defined as the ratio of fuel energy to applied electrical power, calculated using the following equation: $E_{NH3} = ((1.23 - E^0_{NH3}) FE_{NH3})/(1.23 - E)$, where $E^0_{NH3}$ represents the equilibrium potential of nitrate electroreduction to ammonia, which is 0.69 V[70]. $FE_{NH3}$ is the Faradaic efficiency for ammonia. 1.23 V is the equilibrium potential of water oxidation (i.e. assuming the overpotential of the water oxidation is zero). $E$ is the applied potential vs. RHE after 80% iR correction.

## $^{15}$NO$_3^-$ isotope labelling experiments and $^{14}$NH$_3$ quantification by $^1$H NMR

To determine the $^{14}$NH$_4^+$ yield after 1 h electrolysis of −0.1 mol L$^{-1}$ Na$^{14}$NO$_3$ at −0.1 V (vs. RHE) a calibration curve of $^1$H NMR (400 MHz) measurements was made using a series of standard $^{14}$NH$_4$Cl solutions. Typically, 125 μL of the standard solution or post-electrolysis electrolyte were mixed with 125 μL 15 mmol L$^{-1}$ maleic acid in DMSO-D6, 50 μL of 4 mol L$^{-1}$ H$_2$SO$_4$ in DMSO-D6 and 750 μL DMSO-D6. The peak area integral ratio of $^{14}$NH$_4^+$ to maleic acid is positively correlated with the concentration of $^{14}$NH$_4^+$. To determine the source of NH$_3$, 0.1 mol L$^{-1}$ Na$^{15}$NO$_3$ (>98 atom% $^{15}$N, ≥99% purity) in 0.1 mol L$^{-1}$ NaOH was used as the feeding electrolyte for 1 h electrolysis at −0.1 V (vs. RHE) and $^{15}$NH$_4^+$ in the electrolyte was detected by $^1$H NMR.

## Deuterium kinetic isotope effect (KIE)

To determine the KIE, sodium deuteroxide (NaOD) and 99% D$_2$O were used instead of NaOH and water as electrolyte, respectively, at the same concentrations. The KIE values can be calculated according to the equation: $KIE = J_H/J_D$, in which $J_H$ was the NO$_3$RR current density obtained in H$_2$O/NaOH, and $J_D$ represented current densities in D$_2$O/NaOD solution.

## Data availability

The data that support the findings of this study are available within the paper and its Supplementary Information. The data generated in this study are provided in the Source Data file. Source data are provided with this paper.

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

## Acknowledgements

The project has received funding from the European Research Council (ERC) under the European Union's Horizon 2020 research and innovation program (Grant Agreement CasCat [833408] received by W.S.) and from the Deutsche Forschungsgemeinschaft (DFG, German Research Foundation) under Germanys Excellence Strategy—EXC (2033–390677874—RESOLV received by W.S. J. Z. gratefully acknowledges the financial support for his PhD studies from the Chinese Scholarship Council (CSC). This work was supported by the "Center for Solvation Science ZEMOS" funded by the German Federal Ministry of Education and Research BMBF and by the Ministry of Culture and Research of Nord Rhine-Westphalia. ZGH at Ruhr University Bochum is acknowledged for EELS measurement. The authors are grateful to Martin Trautmann for ICP-MS measurements.

## Author contributions

W.S. supervised and conceived the project. J.Z. conceived the project and conducted the electrochemical experiments, TEM, SEM, and UV–Vis and single-entity electrochemistry. T.Q. provided suggestions during the fabrication of single-particle on carbon nanoelectrode assemblies. S.D. and R.Z. performed XPS measurements. Y-T.C. performed EELS measurements. J.R.C.J. performed the ¹H NMR measurements. S.S. carried out the XRD measurements. B.E. performed the DEMS measurements. All authors contributed to the data analysis and discussions. J.Z. and W.S. wrote the paper.

## Funding

## Competing interests

The authors declare no competing interests.
