## [Peer Review File · Nature Communications]

REVIEWER COMMENTS

Reviewer #1 (Remarks to the Author):

This work reports the synthesis, characterization and application to nitrate electroreduction of a bimetallic Co-B/Ru₁₂ electrocatalyst. With the in-situ reconstruction of the catalyst results of ammonium yield and Faradaic efficiency overpass previously reported data at low overpotentials, therefore highlighting the characteristics of the catalyst.

Electrochemical measurements were carried out working with an H-type cell under defined conditions that included 0.1 mol.L⁻¹ NaNO₃; giving that this work aims to contribute to provide sustainable alternatives to ammonia formation, the influence of the initial concentration of nitrate must be analysed; even if the results obtained are promising, the rate of ammonium formation is still too low to be compared with established technologies, mentioned in the introduction section of the work, such as the Haber-Bosch process. Another question is the source of nitrates and the composition of the feed phase. How can this work make a sound contribution to the ammonia market as considered in the justification of the work?

Regarding the electrocatalyst stability, issue of paramount relevance in any application, the work reports data of five chronoamperometric cycles of 2 hours each, however, the stability must be assessed for long running times under conditions of practical relevance.

Although from the point of view of the fundamentals in the electrosynthesis of catalytic materials the work reports interesting information, the focus of the manuscript should be redefined making the difference between the synthesis of new materials with improved characteristics and real/industrial applications, which require experimental conditions far beyond those selected in the present manuscript.

Reviewer #2 (Remarks to the Author):

In this manuscript, the authors reported a series of bimetallic Co-B/Ru electrocatalysts with different Ru atomic ratios for nitrate reduction to ammonia. The Co-B/Ru₁₂ exhibited the optimized electrochemical nitrate reduction performance with NH₃ Faradaic efficiency of 90.0% and NH₃ yield rate of 7.4 mg cm⁻² h⁻¹ at 0 V vs. RHE. The authors attributed the excellent NO₃RR performance to the regulation of absorbed hydrogen coverage, which largely inhibited the competing HER. Moreover, the authors performed single-entity test coupled with operando TEM to unveil the dynamic change of the active sites during electrochemical reduction. However, the mechanism was not fully evidenced by current experiments results, and the specific roles of Co, B and Ru on electrochemical nitrate reduction were not clarified. So the manuscript is not recommended for publication in Nature communications. Some specific comments are as follows.

1.The authors utilized kinetic isotope effect (KIE) measurements to evaluate the proton transfer kinetics in NO₃RR and HER. The KIE of Co-B/Ru and Ru were higher than that of Co-B, indicating a slower proton transfer kinetics, which was contradictory to the better nitrate performance of Co-B/Ru. Such phenomenon should be explained.

2.The authors claimed that Co-B plays an important role in modulating H* coverage to avoid the HER side reaction, while the Co species was dynamically reconstructed during the reaction. Therefore, the role of

Co-B after electrochemical in situ reconstruction should also be investigated, especially the influence of Co leaching on the stability of electrode.

3. Some experimental or DFT simulation evidences are necessary to clarify how Co-B and Ru modulate the hydrogen adsorption, the NO₃⁻ activation and conversion.

4. The authors are suggested to perform the operando electron paramagnetic resonance (EPR) measurements to capture H* at different applied potentials to verify the grand discrepancies of water dissociation ability.

5. The active sites for nitrate adsorption/reduction and H₂O adsorption/reduction should be clarified.

6. In the XPS of Co2p, the corresponding peak area of CoO/Co-B was obviously fluctuated with the increase of Ru content (Fig S5), so the reliability of these data should be re-checked.

7. In Figure 4 c-e, the plateau current in CV curves was adopted as an indicator of the number of active sites of the Co-B/Ru12, which should be supported by some theoretical analyses or literatures.

8. The dissolved concentrations of Co and Ru after the reaction need to be measured.

9. The NO₃RR activity of Co-B/Ru12 should be compared with those of other reported Ru-containing catalysts.

Reviewer #3 (Remarks to the Author):

In this manuscript, Zhang et al. reported a cobalt/ruthenium-based catalyst for the electrochemical conversion of nitrate to ammonia. The authors demonstrated the efficacy of a synergistic effect between the two components in modulating the adsorbed hydrogen to achieve a high FE at a low overpotential. To overcome the limited reaction driving force caused by the low overpotential, an in-situ electrochemical reconstruction was applied, resulting in a high NH₃ yield rate. Furthermore, they employed a single entity electrochemistry coupled with identical location TEM to provide visible evidence of the relationship between the reconstruction process and the catalyst's intrinsic activity change. This is an interesting and thorough paper with a complete supplementary material. I suggest this paper for publication after considering the comments listed below.

1. The authors elucidate the distinction between the electrochemical response of a macro-electrode and a nanoelectrode. They then propose that the plateau current observed on the nanoelectrode is indicative of active sites limiting the current rather than mass transfer limitations. However, the cyclic voltammetry curves for the macroelectrode are absent. To substantiate this assertion, the authors should provide the cyclic voltammetry curves on the macro-electrode.

2. Some prior references regarding *H-facilitated NH₃ formation can be considered (e.g., Adv. Funct. Mater. 2023, 33, 2209843, DOI: 10.1002/adfm.202209843).

2. The author mention that the in-situ reconstruction is induced from Cobalt leaching, I am wondering where is the leached Cobalt going? Is it possible that the dissolved copper ions may be deposited back.

3. It is unclear why the author applied a maximum of 10 CV cycles to a single entity electrochemistry. The reasons behind this decision should be elucidated.

4. In Figure 2a, the author demonstrates the application of a thin catalyst layer on a carbon paper substrate via drop coating. This method is more rational for normalizing the current density by the geometric area compared to the 3D substrate, such as metal foams or other typical porous carbon paper substrates. It is essential for the authors to elucidate the treatment they conducted on the carbon

substrate.

5. In Figure S5b, the peak attributed to CoO in Co-B is absent, but a minor peak for B-Co can be discerned in Figure Sc. Given that the Co-B surface is susceptible to oxidation, it would be suggested to conduct an XPS analysis on the fresh prepared sample (Co-B).

6. In figure 3e, pure Ru show a drastic deactivation process, can the authors provide further discussions and explanations into this?

7. In the ECSA measurements (figure S21f), we see the significantly different Cdl values for Co-B and Ru, I am wondering the particle size distribution of those catalysts.

8. What about the details of accurately selecting and placing the single particles by using the micromanipulator arm inside the SEM?

Reviewer #1 (Remarks to the Author):

This work reports the synthesis, characterization and application to nitrate electroreduction of a bimetallic Co-B/Ru₁₂ electrocatalyst. With the in-situ reconstruction of the catalyst results of ammonium yield and Faradaic efficiency overpass previously reported data at low overpotentials, therefore highlighting the characteristics of the catalyst.

We thank the Reviewer for the positive assessment of our work and the constructive feedback

Electrochemical measurements were carried out working with an H-type cell under defined conditions that included 0.1 mol.L⁻¹ NaNO₃; giving that this work aims to contribute to provide sustainable alternatives to ammonia formation, the influence of the initial concentration of nitrate must be analysed; even if the results obtained are promising, the rate of ammonium formation is still too low to be compared with established technologies, mentioned in the introduction section of the work, such as the Haber-Bosch process.

We followed the reviewer's suggestion. To achieve an even higher ammonia production rate, we performed experiments using different NO₃⁻ concentrations (0.2 M, 0.5 M, and 1 M). As shown in the figures below, the NH₃ yield rate (Y_{NH₃}) significantly increases with higher NO₃⁻ concentrations, reaching the highest yield of 47.58 ± 2.69 mg h⁻¹ cm⁻² (NH₃ partial current around 600 mA cm⁻²) at -0.1 V (vs. RHE) in 0.5 mol L⁻¹ NO₃⁻. From a lab-scale perspective, this performance is promising compared to other reported studies (Table S1; SI). When compared to reported values from lab-scale Haber-Bosch processes at lower pressure, our NH₃ production rate is significantly higher (Table S2, SI). We added corresponding descriptions and Figure 2f and Figure 2g to the manuscript and Figure S16 in the SI.

Changes page 8, manuscript: To pursue the higher performance of NO₃RR, the impact of various initial nitrate concentrations (C_{NO₃⁻}) on the FE_{NH₃} and Y_{NH₃} at Co-B/Ru₁₂ was further investigated. The catalyst shows negligible changes in the FE_{NH₃} when C_{NO₃⁻} increases from 0.1 mol L⁻¹ to 0.2 mol L⁻¹ within the potential window from 0 V to -0.10 V (Fig. 2f). However, the FE_{NH₃} begins to decrease at higher C_{NO₃⁻} levels of 0.5 mol L⁻¹ and 1.0 mol L⁻¹. In terms of Y_{NH₃}, a volcano-type trend is observed with varying C_{NO₃⁻} (Fig 2g). The catalyst achieves its highest Y_{NH₃} of 47.6 mg h⁻¹ cm⁻² and a high J_{NH₃} of 600 mA cm⁻² (Fig. S17, SI) at -0.1 V (vs. RHE) in 0.5 mol L⁻¹. These results indicate that the conversion selectivity and yield do not follow a simple linear correlation with the reactant concentration.

Supplementary Table S1 | Comparison of NH₃ synthesis activity of Co-B/Ru₁₂ with other catalysts for the NO₃RR at ambient conditions.

Catalysts	Electrolyte	Potential (V vs. RHE)	NH ₃ yield rate	FE _{NH₃} (%)	Double layer capacitance	Reference
Co-B/Ru ₁₂ on carbon paper	0.1 mol L ⁻¹ NaOH and 0.1 mol L ⁻¹ NaNO ₃	0.0V	7.4 mg h ⁻¹ cm ⁻²	90.5%	1.16 mF cm ⁻²	This work
Co-B/Ru ₁₂ on carbon paper	0.1 mol L ⁻¹ NaOH and 0.1 mol L ⁻¹ NaNO ₃	-0.2 V	15 mg h ⁻¹ cm ⁻²	87.5%	1.16 mF cm ⁻²	This work
Co-B/Ru ₁₂ on carbon paper	0.1 mol L ⁻¹ NaOH and 0.5 mol L ⁻¹ NaNO ₃	-0.1 V	47.58 mg h ⁻¹ cm ⁻²	83.7%	1.16 mF cm ⁻²	This work
Pd/TiO ₂ on carbon cloth	1 mol L ⁻¹ LiCl and 0.25 mol L ⁻¹ NO ₃ ⁻	-0.7V	1.12 mg h ⁻¹ cm ⁻²	92.1%	5.51 mF cm ⁻²	Energy Environ. Sci. 2021, 14, 3938–3944. ¹
Cu ₂ O+Co ₃ O ₄ on carbon paper	0.1 mol L ⁻¹ NaOH and 0.1 mol L ⁻¹ NaNO ₃	-0.3V	12.76 mg h ⁻¹ cm ⁻²	85.4%	0.84 mF cm ⁻²	Angew. Chem. Int. Ed. 2023, 62, e202214830 ²
PdCu nanocube on carbon paper	1 mol L ⁻¹ KOH and 1 mol L ⁻¹ NO ₃ ⁻	-0.2V	≈1.7 mg h ⁻¹ cm ⁻²	≈72 %	3.59 mF cm ⁻²	Nat. Commun. 2022, 13, 2338 ³
Cu/CuAu ordered SAA on carbon paper	1 mol L ⁻¹ KOH and 1 mol L ⁻¹ NO ₃ ⁻	-0.2V	≈2.72 mg h ⁻¹ cm ⁻²	≈62 %	4.91 mF cm ⁻²	Nat. Synth. 2023, 2, 624 ⁴

Rh@Cu 0.6% on Cu foil	0.1 mol L ⁻¹ KOH and 0.1 mol L ⁻¹ NO ₃ ⁻	-0.2V	≈13.5 mg h ⁻¹ cm ⁻²	93%	N/A	Angew. Chem. Int. Ed. 2022,134, e202202556 ⁵
Co ₃ CuN on carbon paper	0.5 mol L ⁻¹ KOH and 0.032 mol L ⁻¹ NO ₃ ⁻	-0.3V	7.74 mg h ⁻¹ cm ⁻²	97%	N/A	Angew. Chem. Int. Ed. 2023, 62, e202308775 ⁶
Ni ₃₃ /NC-sd on Ti mesh	0.5 mol L ⁻¹ Na ₂ SO ₄ and 0.3 mol L ⁻¹ NO ₃ ⁻	-0.5V	1.2 mg h ⁻¹ cm ⁻²	99%	N/A	Angew. Chem. Int. Ed. 2021, 60, 20711–20716 ⁷
Fe/Ni ₂ P on carbon cloth	0.2 mol L ⁻¹ K ₂ SO ₄ and 0.05 mol L ⁻¹ NO ₃ ⁻	-0.4V	4.16 mg h ⁻¹ cm ⁻²	94.3%	N/A	Adv. Energy Mater. 2022, 12, 2103872 ⁸
Fe SAC on glassy carbon	0.1 mol L ⁻¹ K ₂ SO ₄ and 0.5 mol L ⁻¹ NO ₃ ⁻	-0.66V	≈1.95 mg h ⁻¹ cm ⁻²	75%	N/A	Nat. Commun. 2021, 12, 2870 ⁹
Strained Ru nanoclusters on carbon paper	1 mol L ⁻¹ KOH and 1 mol L ⁻¹ NO ₃ ⁻	-0.2V	19.89 mg h ⁻¹ cm ⁻²	≈100%	0.48 mF cm ⁻²	J. Am. Chem. Soc. 2020, 142, 7036–7046 ¹⁰
Meso-PdN NCs on carbon paper	0.1 mol L ⁻¹ Na ₂ SO ₄ and 0.005 mol L ⁻¹ NO ₃ ⁻	-0.7V	0.376 mg h ⁻¹ cm ⁻²	96%	1.17 mF cm ⁻²	Adv.Mater.2023, 35, 2207305 ¹¹
Ru/Cu ₂ O on Cu foam	1 mol L ⁻¹ KOH and 1 mol L ⁻¹ NO ₃ ⁻	-0.4 V	119 mg h ⁻¹ cm ⁻²	75%	449 mF cm ⁻²	J. Am. Chem. Soc. 2024, 146, 668–676 ¹²
Ru/CuNW on Cu foam	1 mol L ⁻¹ KOH and 0.032 mol L ⁻¹ NO ₃ ⁻	-0.135 V	76.5 mg h ⁻¹ cm ⁻²	96%	740.6 mF cm ⁻²	Nat. Nanotechnol.2022, 17, 757-769 ¹³
Pd ₇ Ru ₂₆ on carbon fiber paper	1 mol L ⁻¹ KOH and 0.032 mol L ⁻¹ NO ₃ ⁻	-0.5 V	20.6 mg h ⁻¹ cm ⁻²	91.6%	110.8 mF cm ⁻²	Chem. Sci., 2024, 15, 8204–8215 ¹⁴
Ru-Fe ₂ O ₃ on carbon cloth	0.5 mol L ⁻¹ Na ₂ SO ₄ and 0.1 mol L ⁻¹ NO ₃ ⁻	-0.9 V	5.6 mg h ⁻¹ cm ⁻²	72.8%	4.74 mF cm ⁻²	Appl. Catal., B.2024, 351,123967 ¹⁵
Ru SASs/Co HNSs on carbon paper	1 mol L ⁻¹ NaOH and 1 mol L ⁻¹ NaNO ₃	-0.3V	8.211 mg h ⁻¹ cm ⁻²	100%	N/A	Chem. Eng. J.; 2024,490, 151883 ¹⁶
Ru SA-NC on carbon paper	1 mol L ⁻¹ KOH and 0.5 mol L ⁻¹ KNO ₃	-0.6 V	2.278 mg h ⁻¹ cm ⁻²	72.8%	N/A	ACS Nano, 2023, 17, 3483-3491 ¹⁷
RuOx/Pd on carbon cloth	1 mol L ⁻¹ KOH and 0.51 mol L ⁻¹ KNO ₃	-0.5 V	23.5 mg h ⁻¹ cm ⁻²	98.6%	N/A	ACS Nano, 2023, 17, 1081-1090 ¹⁸

Supplementary Table S2 | Comparison of the ammonia synthesis rate during NO₃RR of the proposed Co-B/Ru₁₂ with reports from lab-scale Haber-Bosch processes (milder reaction conditions) and with state-of-the-art electrocatalytic and photocatalytic NRR routes.

Ammonia synthesis route	Catalyst	NH ₃ synthesis rate	Operation condition	Reference
Haber-Bosch process	Ba ₂ RuH ₆ /MgO	35 mmol g ⁻¹ _{cat} h ⁻¹	300°C and 10 bar	Nat Catal. 2021,11,959-967 ¹⁹
Haber-Bosch process	Ni/LaN	5.543 mmol g ⁻¹ _{cat} h ⁻¹	400°C and 1 bar	Nature 2020,7816,391-395 ²⁰
Haber Bosch process	Ru/LaCoSi	18.5 mmol g ⁻¹ _{cat} h ⁻¹	400°C and 9 bar	J. Am. Chem. Soc.2022,144,8683-8692 ²¹
N ₂ reduction reaction	Ru single atom catalysts	7.12 mmol g ⁻¹ _{cat} h ⁻¹	25°C; ambient pressure	Adv. Mater. 2018, 30, 1803498 ²²
N ₂ reduction reaction	Proton-filtering covalent organic frameworks	16.89 mmol g ⁻¹ _{cat} h ⁻¹	25°C; ambient pressure	Nat Catal. 2021,4, 322-331 ²³
NO ₃ ⁻ reduction reaction	Co-b/Ru ₁₂	5.597 mmol g ⁻¹ _{cat} h ⁻¹	25°C; ambient pressure	This work

However, we agree with the reviewer's point that for large-scale ammonia production, this performance is still too low to compare with the well-established industrial Haber Bosch process. To clarify this, we rephrased the corresponding description in the introduction part.

Changes page 2, manuscript: The NH₃ yield rate through electrocatalytic NO₃RR is still not comparable to that of the industrial Haber-Bosch process. Therefore, utilizing the full potential of catalysts to further increase the NH₃ yield rate and understanding their reaction mechanisms in NO₃RR is the key to further promote the application of the NO₃RR.

Another question is the source of nitrates and the composition of the feed phase. How can this work make a sound contribution to the ammonia market as considered in the justification of the work?

We agree with the reviewer's point that the source of nitrate for a potential industrial application needs to further be clarified. Despite our contribution is of more fundamental nature and for a practical application many more steps are needed, we rephrased the introduction accordingly.

Changes page 1, 2, manuscript: However, over 96% of NH_3 production relies on the Haber-Bosch process, using H_2 derived from fossil fuels consuming approximately 1% to 2% of the global energy, and contributing to about 1.5% to the worldwide CO_2 emissions. With the NH_3 market expected to expand significantly due to its potential as clean hydrogen-rich but carbon-free fuel or for storing H_2 for transport purposes, alternative green NH_3 synthesis routes need to be considered for achieving climate neutrality by 2050. Electrosynthetic NH_3 formation technologies from nitrogen-containing feedstocks (N_2 , nitrate, nitrite and nitric oxide) emerge as promising alternatives to the energy- and carbon-emission-intensive Haber-Bosch process due to their milder working conditions and compatibility with renewable energy provision. Among those technologies, the electrocatalytic reduction of nitrate (NO_3RR) is more thermodynamically favored for producing NH_3 compared to the direct N_2 reduction reaction (eNRR), owing to the lower dissociation energy of the $\text{N}=\text{O}$ bond (204 kJ mol^{-1}). In addition, nitrate sources with concentrations $\geq 0.1 \text{ mol L}^{-1}$ suitable for industrial-scale ammonia electrosynthesis are widely available in wastewater from the fertilizer industries, metal smelters, and nuclear power plants. Thus, NO_3RR simultaneously represents a promising waste-to-value strategy that contributes to alleviating the global nitrogen cycle imbalance.

Regarding the electrocatalyst stability, issue of paramount relevance in any application, the work reports data of five chronoamperometric cycles of 2 hours each, however, the stability must be assessed for long running times under conditions of practical relevance.

As pointed out before, the manuscript aims on suggesting a novel catalyst system for high current density nitrate to ammonia reduction, however, it was not intended to propose an industrial system at this stage. This would need substantial engineering efforts, electrolyzer optimization, and stability and selectivity tests over thousands of hours. This is not feasible for a laboratory environment and a non-optimized small area electrolyzer. Anyhow, we followed the reviewer's suggestion, and a stability test over a longer period (30 h) in 0.5 M NO_3^- was conducted, as illustrated in the figures below. The catalyst demonstrated a high Y_{NH_3} of $46.1 \text{ mg h}^{-1} \text{ cm}^{-2}$ with a FE_{NH_3} of 85.1% after 15 cycles (30 hours) at -0.1 V (vs. RHE) and maintained a J_{NH_3} of approximately 600 mA cm^{-2} throughout the 30 hours measurement.

We have added a description and the figures below in the manuscript as Figure 2f and Figure S18.

Changes page 8, manuscript: Notably, the catalyst still retains a Y_{NH_3} of $46.1 \text{ mg h}^{-1} \text{ cm}^{-2}$ and a FE_{NH_3} of 85.1% after 15 cycles stability test and maintains a J_{NH_3} of around 600 mA cm^{-2} throughout the 30 h measurements. (Fig. 2h, Fig. S18, SI).

Fig. 2h Chronoamperometric stability test of Co-B/Ru₁₂ at -0.1 V (vs RHE) in 0.5 mol L⁻¹ NO₃⁻, showing the yield rate for NH₃, and the faradaic efficiencies for NH₃ (with electrolyte replacement each 2 h).

Supplementary Fig. S18 | Stability test of Co-B/Ru₁₂. Chronoamperometric stability test of Co-B/Ru₁₂ at -0.1 V (vs RHE) in 0.1 mol L⁻¹ NO₃⁻, showing the partial current for NH₃ (with electrolyte replacement each 2 h).

Although from the point of view of the fundamentals in the electrosynthesis of catalytic materials the work reports interesting information, the focus of the manuscript should be redefined making the difference between the synthesis of new materials with improved characteristics and real/industrial applications, which require experimental conditions far beyond those selected in the present manuscript.

Thanks for the reviewer's suggestion, we have rephrased the corresponding description in the introduction part to make it more focus on the fundamental aspects rather than to compare with the mature industrial Harber-Bosch process. See the changes highlighted in yellow in page 1-2 in manuscript.

Reviewer #2 (Remarks to the Author):

In this manuscript, the authors reported a series of bimetallic Co-B/Ru electrocatalysts with different Ru atomic ratios for nitrate reduction to ammonia. The Co-B/Ru₁₂ exhibited the optimized electrochemical nitrate reduction performance with NH₃ Faradaic efficiency of 90.0% and NH₃ yield rate of 7.4 mg cm⁻² h⁻¹ at 0 V vs. RHE. The authors attributed the excellent NO₃RR performance to the regulation of adsorbed hydrogen coverage, which largely inhibited the competing HER. Moreover, the authors performed single-entity test coupled with operando TEM to unveil the dynamic change of the active sites during electrochemical reduction. However, the mechanism was not fully evidenced by current experiments results, and the specific roles of Co, B and Ru on electrochemical nitrate reduction were not clarified. So the manuscript is not recommended for publication in Nature communications. Some specific comments are as follows.

We thank the reviewer for evaluating our manuscript and appreciate their concerns regarding the need for deeper insights into the specific roles of Co, Ru, and B. This is contradicting the impression of reviewer 1 who was suggesting more experiments into the direction of the potential application.

We want to emphasize that the novelty of our findings lies in utilizing single-entity electrochemistry to reveal the relationship between catalyst reconstruction and intrinsic activity changes for a highly active NO_3^- -reduction catalyst. This insight sheds light on the mechanism of in-situ electrochemical reconstruction during NO_3RR and offers guidance for the rational design of more advanced electrocatalysts. With the revised version, we hope to convince the reviewer that our manuscript and the obtained results meet the standard of Nature Communications.

1. The authors utilized kinetic isotope effect (KIE) measurements to evaluate the proton transfer kinetics in NO_3RR and HER. The KIE of Co-B/Ru and Ru were higher than that of Co-B, indicating a slower proton transfer kinetics, which was contradictory to the better nitrate performance of Co-B/Ru. Such phenomenon should be explained.

In Figure 3c, we state that Co-B/Ru_x and Co-B were higher than that of Ru, we think the reviewer want to ask that the higher KIEs of Co-B/Ru and Co-B indicate a slower transfer kinetic which was contradictory to the better nitrate performance of Co-B/Ru. Of course, we agree that higher KIEs reflect slower proton transfer. However, in alkaline condition (0.1 M NaOH) for NO_3RR , we cannot directly correlate a linear increasing kinetic between the KIE values and the NO_3RR performance, because the KIEs reflect the combined effect of the proton generation and transfer kinetics involved in the water dissociation process with the cleavage of the HO-H bond (the only proton generation step in alkaline solution) and the subsequent $^*\text{H}$ consumption process in NO_3RR or in the competitive reaction of the HER.

Discussion in the water dissociating step: Together with lower HER Tafel slopes (Fig. 3a;b) and lower $^*\text{H}$ coverage values (Fig. 3d) observed on catalysts containing Co-B compared to Ru, the observed KIEs that Co-B plays a significant role in preventing excessive $^*\text{H}$ generation from water dissociation. In other words, the rate of proton generation is slower due to the interaction of Co-B and Ru. This decrease in $^*\text{H}$ generation reduce the recombination of two $^*\text{H}$ atoms to form H_2 , consequently suppressing the HER.

Discussion in the hydrogenation step for NO_3RR step: To clarify this, we further investigated the relationship between the KIE and the partial current of NH_3 at a potential of -0.2 V (vs. RHE) (see figures below; Fig. S26, SI), where both HER and NO_3RR coexist for Ru-containing catalyst. As shown in the figure below, we observed that the KIE on Co-B/Ru_x located between the values for pure Co-B and Ru; however, Co-B/Ru_x shows a higher partial NH_3 current. This indicates an optimized proton transfer rate due to the interaction between Co-B and Ru, suggesting that the generated protons from water dissociation at Co-B/Ru_x can be fast consumed in the hydrogenation step for the NO_3RR rather than for the HER, ultimately contributing to the high NH_3 yield rate.

In addition, a recent study by Yan et al. (<https://doi.org/10.1039/D4SC00558A>) reported similar results, demonstrating that a PdRu alloy has a higher KIE values but still achieves better NO_3RR performance than pure Pd.

For clarification, we have revised the statement in our manuscript and included the corresponding figure in the SI.

Changes page 8, manuscript: Catalysts containing Co-B exhibit significantly higher KIE values compared to Ru within the potential range of 0 V to -0.2 V (vs. RHE), implying that the Co-based catalysts may encounter a larger barrier during the generation of $^*\text{H}$ from water dissociation or in the subsequent transfer of the produced $^*\text{H}$ during the hydrogenation step during the NO_3RR . We investigated the $^*\text{H}$ coverage on

the catalysts by integrating the *H desorption peak area observed in cyclic voltammograms (Fig. S24, SI). No desorption peak was observed for Co-B (inset of Figure S24, SI), indicating that *H is located exclusively on the Ru active sites.³⁵ In Figure 3d, pure Ru exhibits the highest *H coverage of 25.9 mC, which decreases sharply with increasing Co-B content. This decrease remains significant even when normalized by the double layer capacitances (C_{dl}) of the catalysts (Fig. S25, SI). These findings indicate that pure Ru has the fastest *H generation rate from water dissociation, while the presence of Co-B in the bimetallic catalyst can effectively modulate the *H generation rate to avoid the recombination of two *H atoms to form H_2 . Furthermore, as shown in Fig. S26 (SI), the Co-B/Ru_x catalyst exhibits higher NH_3 partial currents with KIE values located between those of Co-B and pure Ru. This suggests an optimized proton transfer rate resulting from the interaction between Co-B and Ru, which favors the requirement of the hydrogenation step for NO_3RR rather than HER.⁶²

Supplementary Fig. S26 | Plot of partial current for NH_3 against the KIE values of different catalysts.

2. The authors claimed that Co-B plays an important role in modulating H^* coverage to avoid the HER side reaction, while the Co species was dynamically reconstructed during the reaction. Therefore, the role of Co-B after electrochemical in situ reconstruction should also be investigated, especially the influence of Co leaching on the stability of electrode.

We understand the reviewer's concern regarding catalyst stability. From our single-entity experiment (see Fig. S39, SI), we observed that the leached Co in the electrolyte redeposits onto the electrode, forming an optimal structure for the catalytic process. Our stability tests showed that the catalyst maintained a relatively stable NH_3 yield rate and FE after both 10 and 30 hours of measurement in 0.1 M and 0.5 M NO_3^- , suggesting that keep their activity over prolonged electrolysis times. We further examined the structural changes of the catalyst after 10 hours using TEM. The reconstructed structure forms a nanosheet matrix of $(Co(OH)_2)$ covered with Ru. A similar structure was also found in the catalyst after the stability tests. To clarify this, we have added the figures below to the SI and descriptions to the manuscript.

Changes page 13, manuscript: The leached cobalt ions can redeposit onto the catalyst or electrode surface from the electrolyte, forming a reconstructed structure (Fig. S39, SI). TEM images of the catalyst on carbon paper after 10 hours stability measurement show that the reconstructed structure forms a nanosheet matrix composed of $Co(OH)_2$ with a coverage of Ru (Fig. S40, SI). A similar structure was observed after extended stability tests (Fig. S41, SI).

Supplementary Fig. S40 | Structure of Co-B/Ru₁₂ after reaction in 0.1 mol L⁻¹ NaNO₃. (a) TEM image of Co-B/Ru₁₂ after 10 hours chronoamperometry at -0.1 V (vs. RHE). (b) HRTEM image of Co-B/Ru₁₂ after 10 hours chronoamperometry at -0.1 V (vs. RHE). (c) STEM image of Co-B/Ru₁₂ after 10 hours chronoamperometry at -0.1 V (vs. RHE). (d) EDX mapping images of Co-B/Ru₁₂ after 10 hours chronoamperometry at -0.1 V (vs. RHE).

Supplementary Fig. S41 | Structure of Co-B/Ru₁₂ after NO₃RR in 0.5 mol L⁻¹ NaNO₃. (a-c) three STEM images and corresponding EDS mapping images of Co-B/Ru₁₂ after 30 hours chronoamperometry at -0.1 V (vs. RHE).

3. Some experimental or DFT simulation evidences are necessary to clarify how Co-B and Ru modulate the hydrogen adsorption, the NO_3^- activation and conversion.

From an experimental perspective, the role of Co-B in Co-B/Ru₁₂ in inhibiting the water dissociation step (related to hydrogen adsorption) has been illustrated through in-situ DEMS and *H coverage measurements. Although we provided experimental evidence on how Co-B inhibits the excessive *H to enhance the NO_3RR , we acknowledge that simulations could further clarify these roles. However, due to the hybrid nature of the catalyst (comprising both crystalline and amorphous components) and the reconstruction process during the reaction, it is almost impossible to design a suitable theoretical model that accurately reflects the real conditions during NO_3RR . At this stage, we are convinced that standard DFT calculations which are not able to address a sufficiently high amount of different surface variability, a sufficient number of explicit water molecules, solvation of all components contributing to the reaction including cations and anions in variable concentrations, the structure and dynamicity of the electrochemical double layer and its changes during electron transfer cannot be adequately described, that we do not see an additional value in adding under-complex theoretical descriptions.

4. The authors are suggested to perform the operando electron paramagnetic resonance (EPR) measurements to capture H^* at different applied potentials to verify the grand discrepancies of water dissociation ability.

We appreciate the reviewer's suggestion to perform operando electron paramagnetic resonance (EPR) measurements to capture H^* at different applied potentials. However, inserting the electrode with the adsorbed catalyst into the EPR cell and applying a potential which is comparable to the conditions in the electrochemical cell concerning separation of the counter electrode compartment, position of the reference electrode we feel that this very unusual experiment will not answer quantitatively the raised questions. Since we experimentally confirmed the significant discrepancies in water dissociation ability on different catalysts through the desorption peak area of *H in the corresponding voltammetric measurements (Fig. S24, SI), we are convinced that the surface coverage of *H is confirmed safely with the already used electrochemical method. We are hence convinced that the proposed operando EPR measurements would not add any additional knowledge while these measurements are prohibitively difficult.

5. The active sites for nitrate adsorption/reduction and H_2O adsorption/reduction should be clarified.

For the H_2O adsorption active sites, we suppose that they are on the Ru active sites, as we did not observe any *H desorption peak in pure Co-B (Fig. S24, SI). For the active sites involved in NO_3^- adsorption, we compared the J_{NH_3} of Co-B and Ru. Ru exhibits a significantly higher J_{NH_3} when normalized by the geometric area at 0 V (vs. RHE) compared to pure Co-B (see Fig. a below), indicating its higher activity in primarily adsorbing NO_3^- . However, when normalized by their corresponding double-layer capacitances, as shown in Fig. b below, the difference becomes smaller at 0 V (vs. RHE). This indicates that Co-B also has a certain intrinsic activity for adsorbing NO_3^- . And its activity increases at more negative potentials, surpassing Ru at potentials below -0.15 V (vs. RHE). These results suggest that both Co-B and Ru adsorb NO_3^- differently before converting it to NH_3 at different potentials. In Fig c below, the J_{NH_3} of Co-B/Ru₁₂ is much higher than that of the individual components, indicating that the effect cannot be simply explained by the individual contribution of the two components. Therefore, we must be very cautious in attributing the active sites responsible for adsorbing NO_3^- and converting it to NH_3 solely to either Ru or Co-B. To clarify this, we have added the figures below to the SI and we modified the manuscript accordingly.

Changes page 8, manuscript: We investigated the *H coverage on the catalysts by integrating the *H desorption peak area observed in the corresponding cyclic voltammograms (Fig. S24, SI). No desorption peak was observed for Co-B (inset of Figure S24, SI), indicating that *H is located exclusively on the Ru active sites.³⁵

Changes page 9, manuscript: To investigate the possible active site for adsorbing NO_3^- , we compared the J_{NH_3} normalized by the C_{dl} for Co-B and Ru (Fig. S27a, SI), indicating that both Co-B and Ru have varying levels to adsorb NO_3^- and convert it to NH_3 at different potentials. Fig. S 27b (SI) demonstrates that the J_{NH_3} of Co-B/Ru₁₂ is much higher than that of the individual components, indicating that the effect cannot be simply explained by the individual contribution of Co-B and Ru. Therefore, attributing the active sites responsible for adsorbing NO_3^- and converting it to NH_3 solely to either Ru or Co-B may be too simple.

Fig. (a) Partial current density of NH_3 of Co-B and Ru normalized by the geometric area of the working electrode. (b) Partial current density of NH_3 of Co-B and Ru normalized by the double-layer capacitances. (c) Partial current density of NH_3 of Co-B and Ru normalized by the double-layer capacitances.

6. In the XPS of Co2p, the corresponding peak area of $\text{Co}^0/\text{Co-B}$ was obviously fluctuated with the increase of Ru content (Fig S5), so the reliability of these data should be re-checked.

We thank the reviewers for pointing out the differences in peak area of $\text{Co}^0/\text{Co-B}$ with the increase of Ru. This may be caused by surface oxidation of the catalyst, as we cannot ensure a completely O_2 -free environment during sample storage and delivery process to the XPS setup. To avoid any misunderstandings, we have removed the high-resolution XPS spectra in Fig. S5 and the corresponding description in the manuscript.

7. In Figure 4 c-e, the plateau current in CV curves was adopted as an indicator of the number of active sites of the Co-B/Ru₁₂, which should be supported by some theoretical analyses or literatures.

We followed the reviewer's comment, we cited the publications below in the field of single particle on nanoelectrode to support the description and to make it clearer, we added further discussion in the manuscript on page 12.

Zhou, M., Bao, S. & Bard, A. J. Probing Size and Substrate Effects on the Hydrogen Evolution Reaction by Single Isolated Pt Atoms, Atomic Clusters, and Nanoparticles. *J. Am. Chem. Soc.* 141, 7327–7332 (2019).

Kim, J. & Bard, A. J. Electrodeposition of Single Nanometer-Size Pt Nanoparticles at a Tunneling Ultramicroelectrode and Determination of Fast Heterogeneous Kinetics for $\text{Ru}(\text{NH}_3)_6^{3+}$ Reduction. *J. Am. Chem. Soc.* 138, 975–979 (2016).

Jin, Z. & Bard, A. J. Atom-by-atom electrodeposition of single isolated cobalt oxide molecules and clusters for studying the oxygen evolution reaction. *PNAS* 117, 12651–12656 (2020).

Changes page 110, manuscript: The ultra-fast mass transfer toward the nanoelectrode can rule out that the plateau is caused by diffusion limitations. Instead, it is due to the maximum turnover of the fully occupied active sites on the Co-B/Ru₁₂ particle, which is widely used to precisely estimate the electrochemically active size for a given reaction on nanoelectrodes.⁶⁷⁻⁶⁹

8. The dissolved concentrations of Co and Ru after the reaction need to be measured.

We followed the reviewer's suggestion and conducted ICP-MS measurements after electrolysis. The concentrations of cobalt and ruthenium in the electrolyte after 1 hour of electrolysis under NO₃RR at different potentials are shown in the figures below. To clarify this, we added the corresponding description in the manuscript and added the figure into the SI.

Supplementary Fig. S38 | Co and Ru concentration determined by ICP-MS in 1-hour post-electrolyzed electrolyte in 0.5 mol L⁻¹ NaNO₃ at different potentials on Co-B/Ru₁₂ modified carbon paper. Error bars denote the standard deviations from at least three independent measurements.

Changes page 12, manuscript: This suggests that Co leaching could be responsible for inducing the structural reconstruction of the catalyst, which is further evidenced by the much higher concentration of Co compared to Ru in the electrolyte after 1 hour of electrolysis at different potentials (Fig. S38, SI).

9. The NO₃RR activity of Co-B/Ru₁₂ should be compared with those of other reported Ru-containing catalysts.

We followed the reviewers suggestions, and we further compared our results with other recent published Ru containing catalyst. The following table was inserted to the SI.

Catalysts	Electrolyte	Potential (V vs. RHE)	NH ₃ yield rate	FE _{NH3} (%)	Double layer capacitance	Reference
Co-B/Ru ₁₂ on carbon paper	0.1 mol L ⁻¹ NaOH and 0.1 mol L ⁻¹ NaNO ₃	0.0V	7.4 mg h ⁻¹ cm ⁻²	90.5%	1.16 mF cm ⁻²	This work
Co-B/Ru ₁₂ on carbon paper	0.1 mol L ⁻¹ NaOH and 0.1 mol L ⁻¹ NaNO ₃	-0.2 V	15 mg h ⁻¹ cm ⁻²	87.5%	1.16 mF cm ⁻²	This work

Co-B/Ru ₁₂ on carbon paper	0.1 mol L ⁻¹ NaOH and 0.5 mol L ⁻¹ NaNO ₃	-0.1 V	47.58 mg h ⁻¹ cm ⁻²	83.7%	1.16 mF cm ⁻²	This work
Pd/TiO ₂ on carbon cloth	1 mol L ⁻¹ LiCl and 0.25 mol L ⁻¹ NO ₃ ⁻	-0.7V	1.12 mg h ⁻¹ cm ⁻²	92.1%	5.51 mF cm ⁻²	Energy Environ. Sci. 2021, 14, 3938–3944. ¹
Cu ₂ O+Co ₃ O ₄ on carbon paper	0.1 mol L ⁻¹ NaOH and 0.1 mol L ⁻¹ NaNO ₃	-0.3V	12.76 mg h ⁻¹ cm ⁻²	85.4%	0.84 mF cm ⁻²	Angew. Chem. Int. Ed. 2023, 62, e202214830 ²
PdCu nanocube on carbon paper	1 mol L ⁻¹ KOH and 1 mol L ⁻¹ NO ₃ ⁻	-0.2V	≈1.7 mg h ⁻¹ cm ⁻²	≈72 %	3.59 mF cm ⁻²	Nat. Commun. 2022, 13, 2338 ³
Cu/CuAu ordered SAA on carbon paper	1 mol L ⁻¹ KOH and 1 mol L ⁻¹ NO ₃ ⁻	-0.2V	≈2.72 mg h ⁻¹ cm ⁻²	≈62 %	4.91 mF cm ⁻²	Nat. Synth. 2023, 2, 624 ⁴
Rh@Cu 0.6% on Cu foil	0.1 mol L ⁻¹ KOH and 0.1 mol L ⁻¹ NO ₃ ⁻	-0.2V	≈13.5 mg h ⁻¹ cm ⁻²	93%	N/A	Angew. Chem. Int. Ed. 2022,134, e202202556 ⁵
Co ₃ CuN on carbon paper	0.5 mol L ⁻¹ KOH and 0.032 mol L ⁻¹ NO ₃ ⁻	-0.3V	7.74 mg h ⁻¹ cm ⁻²	97%	N/A	Angew. Chem. Int. Ed. 2023, 62, e202308775 ⁶
Ni ₃₅ /NC-sd on Ti mesh	0.5 mol L ⁻¹ Na ₂ SO ₄ and 0.3 mol L ⁻¹ NO ₃ ⁻	-0.5V	1.2 mg h ⁻¹ cm ⁻²	99%	N/A	Angew. Chem. Int. Ed. 2021, 60, 20711–20716 ⁷
Fe/Ni ₂ P on carbon cloth	0.2 mol L ⁻¹ K ₂ SO ₄ and 0.05 mol L ⁻¹ NO ₃ ⁻	-0.4V	4.16 mg h ⁻¹ cm ⁻²	94.3%	N/A	Adv. Energy Mater. 2022, 12, 2103872 ⁸
Fe SAC on glassy carbon	0.1 mol L ⁻¹ K ₂ SO ₄ and 0.5 mol L ⁻¹ NO ₃ ⁻	-0.66V	≈1.95 mg h ⁻¹ cm ⁻²	75%	N/A	Nat. Commun. 2021, 12, 2870 ⁹
Strained Ru nanoclusters on carbon paper	1 mol L ⁻¹ KOH and 1 mol L ⁻¹ NO ₃ ⁻	-0.2V	19.89 mg h ⁻¹ cm ⁻²	≈100%	0.48 mF cm ⁻²	J. Am. Chem. Soc. 2020, 142, 7036–7046 ¹⁰
Meso-PdN NCs on carbon paper	0.1 mol L ⁻¹ Na ₂ SO ₄ and 0.005 mol L ⁻¹ NO ₃ ⁻	-0.7V	0.376 mg h ⁻¹ cm ⁻²	96%	1.17 mF cm ⁻²	Adv.Mater.2023, 35, 2207305 ¹¹
Ru/Cu ₂ O on Cu foam	1 mol L ⁻¹ KOH and 1 mol L ⁻¹ NO ₃ ⁻	-0.4 V	119 mg h ⁻¹ cm ⁻²	75%	449 mF cm ⁻²	J. Am. Chem. Soc. 2024, 146, 668–676 ¹²
Ru/CuNW on Cu foam	1 mol L ⁻¹ KOH and 0.032 mol L ⁻¹ NO ₃ ⁻	-0.135 V	76.5 mg h ⁻¹ cm ⁻²	96%	740.6 mF cm ⁻²	Nat. Nanotechnol.2022, 17, 757-769 ¹³
Pd ₇₄ Ru ₂₆ on carbon fiber paper	1 mol L ⁻¹ KOH and 0.032 mol L ⁻¹ NO ₃ ⁻	-0.5 V	20.6 mg h ⁻¹ cm ⁻²	91.6%	110.8 mF cm ⁻²	Chem. Sci., 2024, 15, 8204–8215 ¹⁴
Ru-Fe ₂ O ₃ on carbon cloth	0.5 mol L ⁻¹ Na ₂ SO ₄ and 0.1 mol L ⁻¹ NO ₃ ⁻	-0.9 V	5.6 mg h ⁻¹ cm ⁻²	72:8%	4.74 mF cm ⁻²	Appl. Catal., B,2024, 351,123967 ¹⁵
Ru SASs/Co HNSs on carbon paper	1 mol L ⁻¹ NaOH and 1 mol L ⁻¹ NaNO ₃	-0.3V	8.211 mg h ⁻¹ cm ⁻²	100%	N/A	Chem. Eng. J.; 2024,490, 151883 ¹⁶
Ru SA-NC on carbon paper	1 mol L ⁻¹ KOH and 0.5 mol L ⁻¹ KNO ₃	-0.6 V	2.278 mg h ⁻¹ cm ⁻²	72.8%	N/A	ACS Nano, 2023, 17, 3483-3491 ¹⁷
RuOx/Pd on carbon cloth	1 mol L ⁻¹ KOH and 0.51 mol L ⁻¹ KNO ₃	-0.5 V	23.5 mg h ⁻¹ cm ⁻²	98.6%	N/A	ACS Nano, 2023, 17, 1081-1090 ¹⁸

Reviewer #3 (Remarks to the Author):

In this manuscript, Zhang et al. reported a cobalt/ruthenium-based catalyst for the electrochemical conversion of nitrate to ammonia. The authors demonstrated the efficacy of a synergistic effect between the two components in modulating the adsorbed hydrogen to achieve a high FE at a low overpotential. To overcome the limited reaction driving force caused by the low overpotential, an in-situ electrochemical reconstruction was applied, resulting in a high NH₃ yield rate. Furthermore, they employed a single entity electrochemistry coupled with identical location TEM to provide visible evidence of the relationship between the reconstruction process and the catalyst's intrinsic activity change. This is an interesting and thorough paper with a complete supplementary material. I suggest this paper for publication after considering the comments listed below.

We highly appreciate the reviewer's positive comments on our work, and we are happy that the reviewer is in favor of our contribution and he/she principally recommends publication in Nature Communications.

1. The authors elucidate the distinction between the electrochemical response of a macro-electrode and a nanoelectrode. They then propose that the plateau current observed on the nanoelectrode is indicative of active sites limiting the current rather than mass transfer limitations. However, the cyclic voltammery curves for the macroelectrode are absent. To substantiate this assertion, the authors should provide the cyclic voltammery curves on the macro-electrode.

We followed the reviewer's suggestion and performed three independent cyclic voltammery measurements on a catalyst (Co-B/Ru₁₂) modified carbon paper substrate at the same scan rate (50 mV s⁻¹) used for the single particle on the nanoelectrode. On the macro-electrode, we expect that the active site-limited current is difficult to observe since the number of particles on the macro-electrode is sufficient to ensure that the reaction continues without being limited by the active sites in the given potential window. As shown in the figures below, no plateau current was observed on the macro-electrode within the provided potential window, supporting our statement. We have added the figure to the supplementary manuscript (Fig. S33, SI) and made the corresponding modifications in the main manuscript. Additionally, we have cited relevant literatures (69-71) in the field of single-entity electrochemistry to theoretically support our statement.

Supplementary Fig. S33 | Three independent CVs of Co-B/Ru₁₂ in 0.1 mol L⁻¹ NaOH and 0.1 mol L⁻¹ NaNO₃ with a scan rate of 50 mV s⁻¹.

2. Some prior references regarding *H-facilitated NH₃ formation can be considered (e.g., Adv. Funct. Mater. 2023, 33, 2209843, DOI: 10.1002/adfm.202209843).

We followed the reviewer's comment and cited the reference in our manuscript (Reference 30)

3. The author mention that the in-situ reconstruction is induced from Cobalt leaching, I am wondering where is the leached Cobalt going? Is it possible that the dissolved copper ions may be deposited back.

We appreciate the reviewer's insightful comments, which provide valuable information regarding the reconstruction process. As shown in the figure below, we evaluated the EDS mapping of a single particle on the carbon nanoelectrode (CNE) after 10 CV cycles. The results clearly indicate the presence of a deposited cobalt layer on the carbon nanoelectrode surface, suggesting that the leached cobalt can be redeposited on the CNE surface. For clarification, we added the corresponding description in the manuscript and added the figure below to the SI (Fig. S39, SI).

Added description in manuscript page 12:

Changes page 12, manuscript: The leached cobalt ions can redeposit onto the catalyst or electrode surface from the electrolyte, forming a reconstructed structure (Fig. S39, SI).

Supplementary Fig. S39 | Ru evolution of CNE-2@Co-B/Ru₁₂ at different CV cycles. EDS mapping images of Co of CNE-2@Co-B/Ru₁₂ before (a), after the 5th CV cycle (b), and after the 10th CV cycle (c).

4. It is unclear why the author applied a maximum of 10 CV cycles to a single entity electrochemistry. The reasons behind this decision should be elucidated.

We conducted a maximum of 10 CV cycles based on the following considerations: due to the much faster reaction speed on a single particle on the CNE, we aimed to observe the gradual catalyst dynamic evolution during the reaction. Based on our experience in single-particle electrochemistry, 10 CV cycles were sufficient to observe the reconstruction process in detail. Moreover, as shown in Figure S34, the rate of increase in active sites slowed down after the 8th CV cycle, suggesting that the reconstruction process reached a limiting level. From a technical perspective, applying too many CV cycles will also increase the likelihood of single particles detaching from the carbon nanoelectrode. To clarify this, we have added a sentence and cited the corresponding literatures.

Changes page 10, manuscript: A maximum of 10 CV cycles was applied to a single particle due to the much faster reaction rate and the speed of structural evolution compared to macroelectrodes, as well as to decrease the likelihood of single particles detaching from the carbon nanoelectrode.^{55,57}

5. In Figure 2a, the author demonstrates the application of a thin catalyst layer on a carbon paper substrate via drop coating. This method is more rational for normalizing the current density by the geometric area compared to the 3D substrate, such as metal foams or other typical porous carbon paper substrates. It is essential for the authors to elucidate the treatment they conducted on the carbon substrate.

We appreciate the positive comment regarding the rational of our choice of the working electrode material. The carbon paper surface was covered with a dense carbon powder layer mixed with PTFE, forming a relatively flat and dense surface. To provide detailed information, we used EDS mapping to investigate the side view of the carbon paper substrate. To clarify this, we have added the corresponding figure to the supplementary manuscript and added the corresponding description to the manuscript.

Supplementary Fig. S6 | SEM image and corresponding EDS mapping of C and F (PTFE).

Changes page 6, manuscript: Owing to the relatively dense surface structure of the carbon paper (Fig. S6, SI), the drop-coated catalyst was able to uniformly form a 2D film with a thickness of around 5 μm (Fig. 2a, and Fig. S7, SI).

6. In Figure S5b, the peak attributed to Co0 in Co-B is absent, but a minor peak for B-Co can be discerned in Figure S5c. Given that the Co-B surface is susceptible to oxidation, it would be suggested to conduct an XPS analysis on the fresh prepared sample (Co-B).

We thank the reviewers for pointing this out. As we cannot ensure a completely O_2 -free environment during the sample storage and delivery process to the XPS setup. To avoid possible misunderstandings, we have removed the high-resolution XPS spectra in Fig. S5 and the corresponding text in the manuscript.

7. In figure 3e, pure Ru show a drastic deactivation process, can the authors provide further discussions and explanations into this?

We consider that the deactivation is primarily due to the accumulation of adsorbed hydrogen on the Ru surface. The amount of hydrogen generated is significantly larger than that consumed during the NO_3RR hydrogenation process. As the reaction proceeds, the excess adsorbed hydrogen that is not promptly consumed continues to accumulate. This accumulation covers the active sites and hinders the adsorption of NO_3^- , ultimately leading to deactivation observed in chronoamperometry measurements. This phenomenon, where adsorbed hydrogen causes deactivation, is also known in nitrate reduction on Pt. To clarify this, we have added the corresponding descriptions and cited the following reference.

<https://www.sciencedirect.com/science/article/pii/S002207289285013S>

<https://pubs.acs.org/doi/10.1021/cr8003696>

Changes page 9, manuscript: Whereas Ru is undergoing a deactivation process (Fig. 3e) possibly due to the excessive formation of $^*\text{H}$ on the Ru surface, which cannot be consumed by the hydrogenation step of NO_3RR , covering the active sites and hindering the adsorption and conversion of NO_3^- .^{2,63}

8. In the ECSA measurements (figure S21f), we see the significantly different C_{dl} values for Co-B and Ru, I am wondering the particle size distribution of those catalysts.

We followed the reviewer's suggestion and performed TEM imaging to compare the size distribution of Co-B and Ru. As shown in the Fig. S2, there is a significant difference in the size distribution of Co-B, Ru, and Co-B/Ru₁₂. The particle size of Co-B is primarily around 50 nm, which is much larger than that of Ru (3.5 nm) and Co-B/Ru₁₂ (7 nm). This difference in particle size corresponds to the difference in the C_{dl} values. We have added the following figures to the SI and included the corresponding text to the manuscript.

Changes page 3, manuscript: Furthermore, as shown in the transmission electron microscopy (TEM) images (Fig. S2, SI), the Ru-containing catalyst exhibits significantly smaller particle sizes compared to pure Co-B, which may provide a higher electrochemical surface area during the NO_3RR .

Supplementary Fig. S2 | Particle size distribution of the catalysts. (a) TEM images of Co-B and the corresponding size distribution curve. (b) TEM images of Co-B/Ru₁₂ and the corresponding size distribution curve. (c) TEM images of Ru and the corresponding size distribution curve.

9. What about the details of accurately selecting and placing the single particles by using the micromanipulator arm inside the SEM?

As shown in Fig. S29 (SI) below, the accurate selection and placement of particles is achieved using a micromanipulator; the movement distance of micromanipulator tip can be adjusted to 10 nm by a controller, allowing for precise selection and placement of single particles in the SEM chamber. To provide a detailed information of this process, we recorded SEM images at each step and included photographs of the operation stages. For clarity, we have added the figure below to the supplementary manuscript and updated the description in the methods section accordingly.

Supplementary Fig. S30 | Process for picking up, transferring, and placing single particle on carbon nanoelectrode. (a) Photograph of the micromanipulator set-up installed in the SEM chamber. (b) SEM images of details steps of placing single particle on carbon nanoelectrode.

REVIEWER COMMENTS

Reviewer #2 (Remarks to the Author):

Despite the considerable efforts has been made to improve the manuscript, several important issues have not been satisfactorily addressed. Therefore, I still believe that the work is not suitable for publication.

1. The Co-B component undergoes significant electrochemical reconstruction to form $\text{Co}(\text{OH})_2$, indicating the instability of the Co-B component. This could not explain why Co-B/Ru12 still possessed a stable NH_3 production reactivity.
2. From the characterization results, B was distributed in the Co-B/Ru12 catalyst. However, the author did not analyze the impact of the B component on NO_3RR .
3. Metal sites in alloy materials generally have more unique electronic structures than single metal sites, and thus affects the adsorption of substrates and intermediates, ultimately influencing the reaction activity. Therefore, it is necessary to provide DFT theoretical calculations to elucidate how Co-B regulates the adsorption of hydrogen and nitrate as well as the subsequent conversion process of nitrate.
4. EPR testing is an important method to verify the H^* generation ability of electrocatalytic materials, which should be provided to support the conclusion.
5. The authors determined the adsorption sites for nitrate and H_2O only with the activity results of individual components (Co-B and Ru), which was inadequate. More rigorous characterization methods must be employed to identify the active sites for accurately understanding the reaction process.

Reviewer #3 (Remarks to the Author):

The authors have addressed most concerns raised by the reviewers and the revised manuscript can be accepted for publication in Nat. Commun.

Reviewer #2 (Remarks to the Author):

Despite the considerable efforts has been made to improve the manuscript, several important issues have not been satisfactorily addressed. Therefore, I still believe that the work is not suitable for publication.

We thank the reviewer for their time and effort in evaluating this manuscript once again. However, we feel that most of the questions raised by the reviewer were already answered in our first revision.

1. The Co-B component undergoes significant electrochemical reconstruction to form Co(OH)_2 , indicating the instability of the Co-B component. This could not explain why Co-B/Ru₁₂ still possessed a stable NH₃ production reactivity.

In fact, we devoted considerable space in the manuscript to describing this reconstruction phenomenon (From Fig. 3e- to Fig. 4 and Fig. 5). By exactly exploiting the instability of Co-B during the reaction, we activate the catalyst, fully exposing as many active sites as possible to achieve higher NH₃ yield rate, which is very important key part in our manuscript. Additionally, we employ single particle electrochemistry combined with identical TEM to visually demonstrate the correlation between the increase in activity and structural reconstruction induced by Co-B.

Furthermore, it is well-established that reconstructed catalysts tend to exhibit greater stability under electrochemical conditions, as they have already adapted to harsh reaction environments (See the references of 46, 47 in the manuscript)

46: Liu, X. et al. Comprehensive Understandings into Complete Reconstruction of Precatalysts: Synthesis, Applications, and Characterizations. *Adv. Mater.* 33, e2007344 (2021).

47: Liu, X. et al. Complete Reconstruction of Hydrate Pre-Catalysts for Ultrastable Water Electrolysis in Industrial-Concentration Alkali Media. *Cell Rep. Phys. Sci.* 1, 100241 (2020).

2. From the characterization results, B was distributed in the Co-B/Ru₁₂ catalyst. However, the author did not analyze the impact of the B component on NO₃RR.

In the ICP-MS measurements shown in the figure below, we observed a high content of B in the electrolyte during the first-hour post-electrolyzed electrolyte, which then maintained a similar content to the blank electrolyte in the following two hours electrolyzed electrolyte. This indicates that B does not redeposit onto the catalyst like Co ions do during NO₃RR (Co forms Co(OH)_2 in the end, Fig. S39b). Therefore, we do not believe that B plays a significant role as an active site contributing to NO₃RR.

Additionally, previous studies in our lab on nickel and cobalt borides indicate that B is not part of the active site but rather contributes to surface reconstruction and high conductivity in the bulk material, which is not in contact with the electrolyte. (See the reference below)

1. Masa, J. et al. Amorphous Cobalt Boride (Co₂B) as a Highly Efficient Nonprecious Catalyst for Electrochemical Water Splitting: Oxygen and Hydrogen Evolution. *Adv. Energy Mater.* 6, 1502313 (2016).

2. Masa, J. et al. Ultrathin High Surface Area Nickel Boride (Ni_xB) Nanosheets as Highly Efficient Electrocatalyst for Oxygen Evolution. *Adv. Energy Mater.* 7, 1700381 (2017).

To clarify this, we have added the corresponding description to the manuscript and modified the figures in the supplementary information (SI).

Added descriptions in page 12 in manuscript:

However, B shows a high concentration in the electrolyte during the initial 1 hour of electrolysis at 0 V (vs. RHE), while maintaining a level similar to the blank electrolyte during subsequent electrolysis at -0.05 V and -0.10 V (vs. RHE). This indicates that B leaches into the electrolyte and does not redeposit onto the catalyst layer to continue contributing to the leaching process as Co does (Fig. S38c, SI). Therefore, the impact of B as a continuous active site in our catalyst is negligible.

Supplementary Fig. S38 | ICP-MS determined elements concentration in 1-hour post-electrolyzed electrolyte in $0.5 \text{ mol L}^{-1} \text{ NaNO}_3$ at different potentials on Co-B/Ru₁₂ modified carbon paper. Concentrations of Co (a), Ru (b), and B (c) at different potentials. After 1 hour of electrolysis at 0 V (vs. RHE), all the post-electrolyzed electrolyte was collected, and the electrochemical cell was cleaned with deionized water before adding fresh electrolyte for the next potential (-0.05 V vs. RHE) electrolysis. The same collection and cleaning procedures were conducted before performing electrolysis at -0.1 V (vs. RHE). The same working electrode was used throughout the 3-stage electrolysis. Error bars denote the standard deviations from three independent measurements.

3. Metal sites in alloy materials generally have more unique electronic structures than single metal sites, and thus affects the adsorption of substrates and intermediates, ultimately influencing the reaction activity. Therefore, it is necessary to provide DFT theoretical calculations to elucidate how Co-B regulates the adsorption of hydrogen and nitrate as well as the subsequent conversion process of nitrate.

In the first revision (R1), we explained why DFT calculations cannot provide deeper insights. Here, we would like to further emphasize that the initial hybrid nature of the catalyst (comprising both crystalline and amorphous components) before the reaction makes it almost impossible to build an accurate model for calculations that reflect the real catalyst under NO₃RR conditions. Additionally, DFT simulations cannot fully incorporate the reconstruction process. More importantly, the novelty of our findings mainly lies in utilizing single-entity electrochemistry and identical location TEM to reveal the relationship between catalyst reconstruction and intrinsic activity changes for a highly active NO₃-reduction catalyst in a visible way.

Question in R1: Some experimental or DFT simulation evidences are necessary to clarify how Co-B and Ru modulate the hydrogen adsorption, the NO₃⁻ activation and conversion.

Our reply in R1: From an experimental perspective, the role of Co-B in Co-B/Ru12 in inhibiting the water dissociation step (related to hydrogen adsorption) has been illustrated through in-situ DEMS and *H coverage measurements. Although we provided experimental evidence on how Co-B inhibits the excessive *H to enhance the NO₃RR, we acknowledge that simulations could further clarify these roles. However, due to the hybrid nature of the catalyst (comprising both crystalline and amorphous components) and the reconstruction process during the reaction, it is almost impossible to design a suitable theoretical model that accurately reflects the real conditions during NO₃RR. At this stage, we are convinced that standard DFT calculations which are not able to address a sufficiently high amount of different surface variability, a sufficient number of explicit water molecules, solvation of all components contributing to the reaction including cations and anions in variable concentrations, the structure and dynamicity of the electrochemical double layer and its changes during electron transfer cannot be adequately described, that we do not see an additional value in adding under-complex theoretical descriptions.

4. EPR testing is an important method to verify the H* generation ability of electrocatalytic materials, which should be provided to support the conclusion.

In our first revision, we explained the reasons for not pursuing EPR from both a technical standpoint and considering whether it would add significant additional knowledge to our work. We want to highlight that we are convinced that the desorption peak of H* on the catalyst, indicated by CV in Fig S24, is a highly effective and direct method to indicate the H* generation ability. This approach has been reliably and widely used for investigating H* generation ability (See the representative reference below).

Bard, A. J., & Faulkner, L. R. (2001). *Electrochemical Methods: Fundamentals and Applications* (2nd ed.). Wiley. (Chapter 6: hydrogen adsorption/desorption and cyclic voltammetry).

Question in R1: The authors are suggested to perform the operando electron paramagnetic resonance (EPR) measurements to capture H* at different applied potentials to verify the grand discrepancies of water dissociation ability.

Our reply in R1: We appreciate the reviewer's suggestion to perform operando electron paramagnetic resonance (EPR) measurements to capture H* at different applied potentials. However, inserting the electrode with the adsorbed catalyst into the EPR cell and applying a potential which is comparable to the conditions in the electrochemical cell concerning separation of the counter electrode compartment, position of the reference electrode we feel that this very unusual experiment will not answer quantitatively the raised questions. Since we experimentally confirmed the significant discrepancies in water dissociation ability on different catalysts through the desorption peak area of *H in the corresponding voltammetric measurements (Fig. S24, SI), we are convinced that the surface coverage of *H is confirmed safely with the already used electrochemical method. We are hence convinced that the proposed operando EPR measurements would not add any additional knowledge while these measurements are prohibitively difficult.

5. The authors determined the adsorption sites for nitrate and H₂O only with the activity results of individual components (Co-B and Ru), which was inadequate. More rigorous characterization methods must be employed to identify the active sites for accurately understanding the reaction process.

First, regarding the determination of the active site for H₂O adsorption, we relied on the desorption peak in the CVs (Fig. S24, SI) rather than activity results. Since we only observed the *H desorption peak on the Ru-containing catalyst, it is supposed to be on the Ru sites. As for the active sites for nitrate adsorption and conversion, we acknowledge that rigorous characterization could involve DFT calculations. However, as discussed in comment 3, we do not believe simulations can accurately reflect the real conditions of our catalyst during NO₃RR.

Question in R1: The active sites for nitrate adsorption/reduction and H₂O adsorption/reduction should be clarified.

Our reply in R1: For the H₂O adsorption active sites, we suppose that they are on the Ru active sites, as we did not observe any *H desorption peak in pure Co-B (Fig. S24, SI). For the active sites involved in NO₃⁻ adsorption, we compared the J_{NH₃} of Co-B and Ru. Ru exhibits a significantly higher J_{NH₃} when normalized by the geometric area at 0 V (vs. RHE) compared to pure Co-B (see Fig. a below), indicating its higher activity in primarily adsorbing NO₃⁻. However, when normalized by their corresponding double-layer capacitances, as shown in Fig. b below, the difference becomes smaller at 0 V (vs. RHE). This indicates that Co-B also has a certain intrinsic activity for adsorbing NO₃⁻. And its activity increases at more negative potentials, surpassing Ru at potentials below -0.15 V (vs. RHE). These results suggest that both Co-B and Ru adsorb NO₃⁻ differently before converting it to NH₃ at different potentials. In Fig c below, the J_{NH₃} of Co-B/Ru₁₂ is much higher than that of the individual components, indicating that the effect cannot be simply explained by the individual contribution of the two components. Therefore, we must be very cautious in attributing the active sites responsible for adsorbing NO₃⁻ and converting it to NH₃ solely to either Ru or Co-B. To clarify this, we have added the figures below to the SI and we modified the manuscript accordingly.

Reviewer #3 (Remarks to the Author):

The authors have addressed most concerns raised by the reviewers and the revised manuscript can be accepted for publication in Nat. Commun.

We highly appreciate the reviewer's recommendation.